# An FGF-driven feed-forward circuit patterns the cardiopharyngeal mesoderm in space and time

**Florian Razy-Krajka, Basile Gravez, Nicole Kaplan, Claudia Racioppi, Wei Wang, Lionel Christiaen***

Center for Developmental Genetics, Department of Biology, College of Arts and Science, New York University, New York, United States

**Abstract** In embryos, multipotent progenitors divide to produce distinct progeny and express their full potential. In vertebrates, multipotent cardiopharyngeal progenitors produce second-heart-field-derived cardiomyocytes, and branchiomeric skeletal head muscles. However, the mechanisms underlying these early fate choices remain largely elusive. The tunicate *Ciona* emerged as an attractive model to study early cardiopharyngeal development at high resolution: through two asymmetric and oriented divisions, defined cardiopharyngeal progenitors produce distinct first and second heart precursors, and pharyngeal muscle (aka atrial siphon muscle, ASM) precursors. Here, we demonstrate that differential FGF-MAPK signaling distinguishes between heart and ASM precursors. We characterize a feed-forward circuit that promotes the successive activations of essential ASM determinants, *Hand-related*, *Tbx1/10* and *Ebf*. Finally, we show that coupling FGF-MAPK restriction and cardiopharyngeal network deployment with cell divisions defines the timing of gene expression and permits the emergence of diverse cell types from multipotent progenitors.
DOI: https://doi.org/10.7554/eLife.29656.001

## Introduction

Developmental genetics knowledge guided progress towards driving mammalian stem cells into forming pure cultures of selected cell types in vitro (e.g. [*Kattman et al., 2011*; *Mazzoni et al., 2011*; *Peljto and Wichterle, 2011*]). By contrast, in the embryo, pluripotent cells generate diverse cell types in defined proportions, as they divide before individual daughter cells adopt distinct fates.

Subsets of the heart and head/neck myocytes recently emerged as related derivatives of multipotent progenitors located in the cardiopharyngeal mesoderm (*Diogo et al., 2015a*; *Tzahor, 2009*; *Tzahor and Evans, 2011*). Early lineage tracing, transplantations and controlled explant culture experiments demonstrated that the anterior splanchnic/pharyngeal mesoderm of amniote embryos can produce either skeletal muscles or heart tissue, depending upon exposure to growth factors and signaling molecules (*Nathan et al., 2008*; *Tirosh-Finkel et al., 2006*; *Tzahor et al., 2003*; *Tzahor and Lassar, 2001*). Clonal analyses in the mouse further revealed the existence of common *Mesp1*-expressing progenitors for subsets of the second heart field-derived cardiomyocytes and branchiomeric facial, jaw, neck and even œsophageal muscles (*Gopalakrishnan et al., 2015*; *Lescroart et al., 2014*; *Lescroart et al., 2015*; *Lescroart et al., 2010*; *Lescroart et al., 2012*). In pluripotent stem cells, controlled *Mesp1* expression can drive mesodermal progenitors towards cardiac and/or skeletal muscle fates (*Bondue et al., 2008*; *Chan et al., 2016*; *Chan et al., 2013*). Proper development of the pharyngeal apparatus and second heart field derivatives require shared inputs from Tbx1, Nkx2.5 and Islet1 transcription factors (e.g. [*Cai et al., 2003*; *George et al., 2015*; *Jerome and Papaioannou, 2001*; *Kelly et al., 2004*; *Merscher et al., 2001*; *Mosimann et al., 2015*; *Nevis et al., 2013*; *Prall et al., 2007*; *Tzahor and Evans, 2011*;

*For correspondence:
lc121@nyu.edu

**Competing interests:** The authors declare that no competing interests exist.

*Vitelli et al., 2002a*; *Watanabe et al., 2012*; *Witzel et al., 2017*; *Yagi et al., 2003*; *Zhang et al., 2006*]). Taken together, this growing body of evidence points to the existence of a mesodermal field of multipotent progenitors capable of producing either SHF-derived cardiomyocytes or branchiomeric skeletal muscles in early vertebrate embryos (*Diogo et al., 2015*; *Mandal et al., 2017*). However, the mechanisms that distinguish fate-restricted heart and head muscle precursors remain largely elusive.

The tunicate Ciona, which is among the closest living relatives to the vertebrates (*Delsuc et al., 2006*; *Putnam et al., 2008*), has emerged as a simple chordate model to characterize multipotent cardiopharyngeal progenitors and the mechanisms that initiate heart vs. pharyngeal muscle fate choices (*Kaplan et al., 2015*; *Razy-Krajka et al., 2014*; *Stolfi et al., 2010*; *Tolkin and Christiaen, 2016*; *Wang et al., 2013*). Ciona tailbud embryos possess two multipotent cardiopharyngeal progenitors on either side. Like their vertebrate counterparts, these cells emerge from *Mesp+* progenitors towards the end of gastrulation; they are induced by FGF-MAPK signaling and have been termed *trunk ventral cells* (aka TVCs; [*Christiaen et al., 2008*; *Davidson and Levine, 2003*; *Davidson et al., 2006*; *Davidson et al., 2005*; *Satou et al., 2004*; *Stolfi et al., 2010*]). TVCs activate conserved cardiac markers, including *Hand*, *Gata4/5/6* and *Nk4/Nkx2-5*, and migrate as polarized pairs of cells, until the left and right pairs meet at the ventral midline and begin to divide asymmetrically along the mediolateral axis (*Figure 1A*; [*Christiaen et al., 2008*; *Davidson et al., 2005*; *Satou et al., 2004*; *Stolfi et al., 2010*]). The first oriented asymmetric divisions produce small median first heart precursors (FHPs), and large lateral second trunk ventral cells (STVCs), which specifically activate *Tbx1/10* (*Davidson et al., 2005*; *Stolfi et al., 2010*; *Wang et al., 2013*). STVCs later divide again to produce small median second heart precursors (SHPs), and large lateral atrial siphon muscle founder cells (ASMFs), which activate *Ebf* (aka *COE*; [*Razy-Krajka et al., 2014*; *Stolfi et al., 2010*; *Stolfi et al., 2015*]). The transcription factors Hand-related (Hand-r)/Notrlc, which is expressed in the TVCs and maintained in the STVCs and ASMFs after each division, and Tbx1/10 are required for *Ebf* activation in the ASMFs, whereas Nk4/Nkx2.5 represses *Tbx1/10* and *Ebf* expression in the second heart precursors (SHPs)(*Razy-Krajka et al., 2014*; *Tolkin and Christiaen, 2016*; *Wang et al., 2013*). Conversely, Tbx1/10 and Ebf inhibit cardiac markers, and likely determinants, such as *Gata4/5/6* and *Hand* (*Razy-Krajka et al., 2014*; *Stolfi et al., 2010*, *2014a*; *Wang et al., 2013*). These regulatory cross-antagonisms underlie the transition from transcriptionally primed multipotent progenitors to separate fate-restricted precursors, by limiting the deployment of the heart- and pharyngeal-muscle-specific programs to their corresponding specific precursors (*Kaplan et al., 2015*).

Here, we identify regulatory mechanisms ensuring the emergence of diverse fate-restricted precursors from multipotent progenitors. We show that differential FGF-MAPK signaling, feed-forward regulatory circuits and coupling with the cell cycle control the spatially restricted activation of *Tbx1/10* and *Ebf*, successively, thus permitting the emergence of both first and second heart precursors, and ASM/pharyngeal muscle precursors from common multipotent progenitors.

## Results

### MAPK signaling is active in the multipotent cardiopharyngeal progenitors and progressively restricted to the pharyngeal muscle precursors

During the earliest stages of cardiopharyngeal development in ascidians, multipotent progenitors display multilineage transcriptional priming, (*Razy-Krajka et al., 2014*; *Stolfi et al., 2014b*), and subsequent regulatory cross-antagonisms segregate distinct cardiopharyngeal programs to their corresponding fate-restricted progenitors (*Stolfi et al., 2010*; *Wang et al., 2013*); reviewed in [*Kaplan et al., 2015*]). For instance, the ASM-specific factor Ebf is necessary and sufficient to terminate the heart program and impose a pharyngeal muscle fate (*Razy-Krajka et al., 2014*; *Stolfi et al., 2010*). However, the symmetry-breaking events leading to cardiopharyngeal mesoderm patterning and ASM-specific expression of *Ebf* remain unknown. We surmised that differential signaling inputs determine the stereotyped spatio-temporal patterning of early cardiopharyngeal progenitors.

The Ciona homologs of specific FGF-MAPK pathway components, including *FGF receptor substrate 2/3* (*Frs2/3*; [*Gotoh et al., 2004*]), *Ets.b*, and *Fgf4/5/6*, are preferentially expressed in the

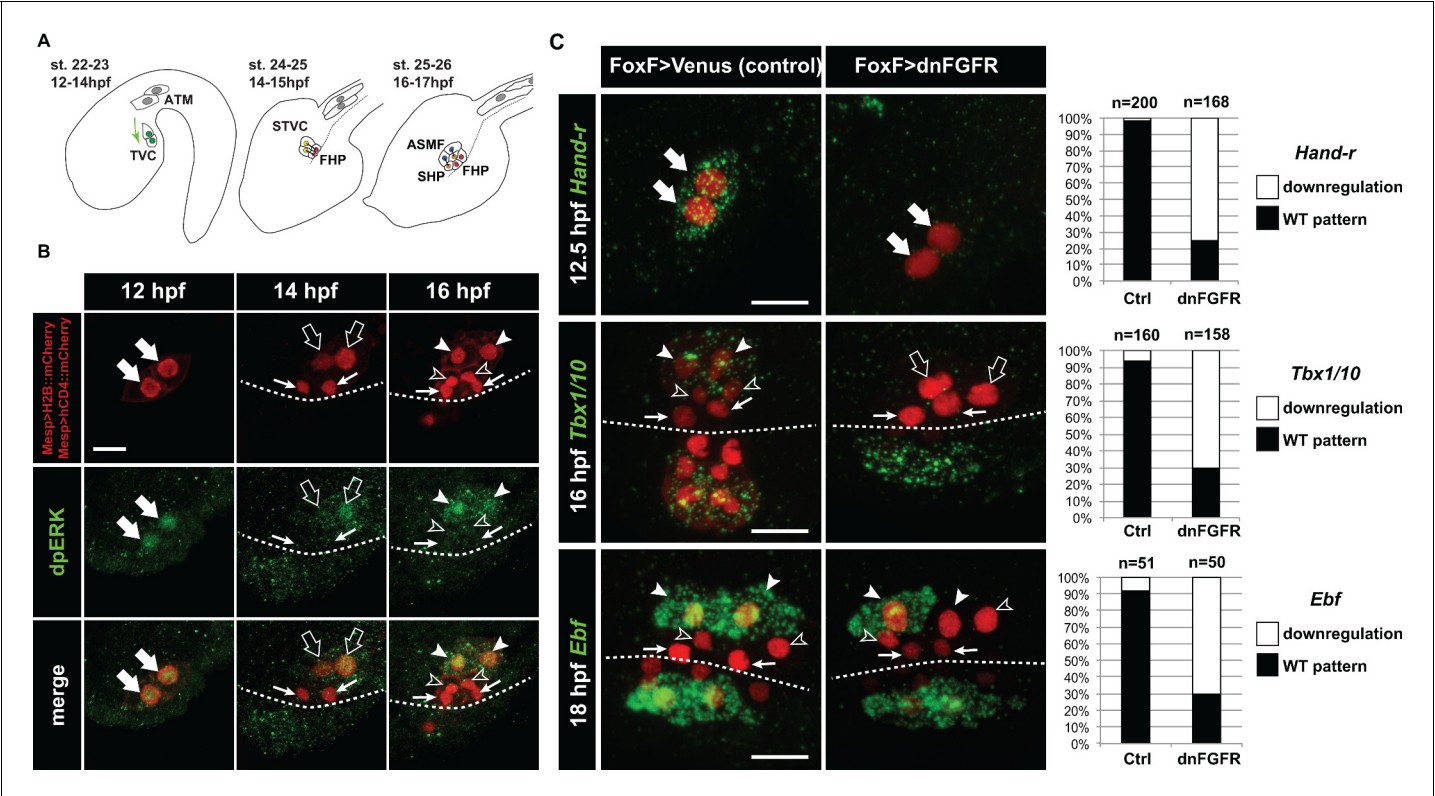

**Figure 1.** Spatio-temporal restriction of ERK activity reflects FGF requirement for the specification of cardiopharyngeal progenitors. (**A**) Schematic of *Ciona* development showing asymmetric cell divisions and resulting cell fates of the cardiopharyngeal mesoderm (CPM). Embryonic and larval stages (St) according to (*Hotta et al., 2007*) with hours post fertilization (hpf) at 18°C. Anterior tail muscle (ATM, gray), trunk ventral cell (TVC, green), secondary TVC (STVC, green), first heart precursor (FHP, red), second heart precursor (SHP, orange), atrial siphon founder cell (ASMF, blue). Black bars link sister cells. Dashed lines: ventral midline. The first stage presents a quasi-lateral view while the second and third stages present quasi-ventral views. Anterior is to the left. Scale bar, 50 μm. (**B**) ERK activity visualized by anti-dpERK antibody (green). TVCs and their progeny are marked by mCherry driven by *Mesp* and revealed by anti-mCherry antibody (red). H$_2$B::mCherry and hCD4::mCherry accumulate in the nuclei and at the cell membrane, respectively. Arrowheads indicate STVCs and ASMFs at 14 and 16 hpf, respectively. Arrows indicate FHPs and open arrowheads mark SHPs. Anterior to the left. Scale bar, 10 μm. See also *Figure 1—figure supplement 1* for broader time series of dpERK immunostaining in the B7.5 lineage. (**C, D**) TVC-specific overexpression of dnFGFR induces loss of expression of key lateral CPM markers visualized by in situ hybridization. (**C**) Representative expression patterns of key CPM genes (*Hand-related*, *Tbx1/10*, *Ebf*) in control embryos (Ctrl, electroporated with Foxf(TVC):bpFOG-1>Venus) and TVC-specific dnFGFR expression (electroporated with *Foxf(TVC):bpFOG-1>dnFGFR::mCherry*) individuals. TVCs and progeny are marked with *Mesp > NLS::lacZ* (red). Loss of expression in half of the TVC progeny, as presented for *Ebf*, is assumed to be due to left-right mosaicism. Arrowheads mark the ASMFs. Anterior is to the left. Scale bar, 10 μm. (**D**) Corresponding histograms with the phenotype proportions. For simplicity, loss of gene expression in half or all of the TVCs and their progeny were combined in the same category. 'n' corresponds to the number of individual halves documented per condition.

DOI: https://doi.org/10.7554/eLife.29656.002

The following figure supplements are available for figure 1:

**Figure supplement 1.** Detailed patterns of MAPK activity during early cardiopharyngeal development.

DOI: https://doi.org/10.7554/eLife.29656.003

**Figure supplement 2.** Late TVC-specific inhibition of FGF-MAPK signaling does not alter early TVC induction .

DOI: https://doi.org/10.7554/eLife.29656.004

TVCs, in the STVCs and in the ASMFs as cells transition from multipotent progenitor to distinct heart vs. ASM fate-restricted states (*Razy-Krajka et al., 2014*). These patterned expressions of MAPK effector genes prompted us to evaluate a role for FGF-MAPK pathway in cardiopharyngeal fate decisions.

We first used an antibody specific to the dual phosphorylated form of Extracellular Regulated Kinase (dpERK) to monitor Mitogen Activated Protein Kinase (MAPK) activity in the cardiopharyngeal mesoderm. We detected dpERK staining in the newly born TVCs, marked by the B7.5-lineage-

specific *Mesp >H2B::mCherry* transgene (*Figure 1—figure supplement 1*), as previously observed (*Davidson et al., 2006*). We also detected weaker but persistent dpERK staining in the TVCs during migration (*Figure 1* and *Figure 1—figure supplement 1*). Following the first and second asymmetric divisions of the TVCs and STVCs, dpERK staining was successively restricted to the more lateral STVCs and ASMFs, respectively (*Figure 1A,B*; *Figure 1—figure supplement 1*).

## The canonical FGF/Ras/MEK/ERK pathway is necessary and sufficient to promote pharyngeal muscle specification in the cardiopharyngeal lineage

This exclusion of MAPK activity from the medial first and second heart precursors opened the possibility that differential ERK activity is required for proper STVC and ASMF vs. heart precursors fate decisions. In Ciona, signaling through the sole FGF receptor (FGFR) governs ERK activity in several developmental processes, including neural induction (*Bertrand et al., 2003*; *Hudson et al., 2003*) and central nervous system patterning (*Haupaix et al., 2014*; *Racioppi et al., 2014*; *Stolfi et al., 2011*; *Wagner et al., 2014*), early endomesoderm and notochord fate specification (*Imai et al., 2002*; *Picco et al., 2007*; *Shi and Levine, 2008*; *Shi et al., 2009*; *Yasuo and Hudson, 2007*). Notably, FGF-MAPK signaling is active in the *Mesp*+ cardiogenic B7.5 blastomeres (*Imai et al., 2006*; *Shi and Levine, 2008*), where targeted misexpression of a dominant negative form of FGFR (dnFGFR) blocks TVC induction (*Christiaen et al., 2008*; *Davidson et al., 2006*). We used a TVC-specific *Foxf* enhancer (*Foxf(TVC):bpFog-1>dnFGFR::mCherry*, hereafter called *Foxf>dnFGFR*; [*Beh et al., 2007*]), to bypass early effects and achieve later misexpression of dnFGFR in the TVCs and their progeny. Importantly, although TVC fate specification and the onset of *Foxf* expression require FGF-MAPK signaling (*Beh et al., 2007*; *Davidson et al., 2006*), we confirmed that this perturbation altered neither initial TVC induction, nor the expression of the *Foxf* driver (*Figure 1—figure supplement 2*). Consistent with proper TVC induction, *Foxf>dnFGFR* prevented neither TVC migration nor asymmetric divisions, but it abolished the expression of both *Tbx1/10* in the STVCs and *Ebf* in the ASMFs (*Figure 1C*). This data indicate that FGF-MAPK signaling is required in the cardiopharyngeal progenitors and/or their progeny for ASM fate specification, beyond the initial TVC induction.

Upon FGF-MAPK-dependent induction, the TVCs express *Hand-related/Hand-r* (renamed after *Notrlc/Hand-like*; [*Christiaen et al., 2008*; *Davidson and Levine, 2003*; *Davidson et al., 2006*; *Satou et al., 2004*; *Stolfi et al., 2015*; *Woznica et al., 2012*]), which encodes a basic helix-loop-helix (bHLH) transcription factor necessary for *Ebf* expression in the ASMFs (*Razy-Krajka et al., 2014*). Moreover, the *Hand-r* TVC enhancer contains putative Ets1/2 binding sites, which are necessary for reporter gene expression, and presumably mediate the transcriptional inputs of FGF-MAPK (*Woznica et al., 2012*). Since *Hand-r* and *Foxf* expressions start at approximately the same time in newborn TVCs, we used *Foxf>dnFGFR*, which did not alter the onset of *Hand-r* expression in the TVCs (*Figure 1—figure supplement 2*), to test whether the maintenance of *Hand-r* expression in migratory TVCs requires prolonged FGF-MAPK inputs. *Foxf>dnFGFR* inhibited *Hand-r* expression in late TVCs (*Figure 1C*), indicating that sustained *Hand-r* expression requires continuous FGF-MAPK signaling, as did TVC-expressed FGF-MAPK pathway components (*Figure 1—figure supplement 2*).

To test whether the spatial restriction of MAPK activity explains the patterned expressions of *Hand-r*, *Tbx1/10* and *Ebf* following asymmetric cell divisions, we used gain-of-function perturbations to force FGF-MAPK activity throughout the cardiopharyngeal mesoderm and assayed gene expression (*Figure 2*). We focused on the canonical FGF-MAPK pathway where signal transduction involves Ras, Raf, MEK and ERK downstream of FGFR and upstream of transcriptional effectors (*Lemmon and Schlessinger, 2010*). We first used M-Ras^G22V, a defined constitutively active form of M-Ras, which mediates FGF signaling in *Ciona*, where other classical *Ras* genes are missing (*Keduka et al., 2009*), and focused on *Htr7* and *Tbx1/10* expression following the first asymmetric TVC division in 15 hours post-fertilization (hpf) embryos. *Htr7* encodes a *trans*-membrane G-protein coupled receptor and, like *Hand-r*, its expression and maintenance in the TVCs require MAPK activity (*Figure 1—figure supplement 2*; [*Razy-Krajka et al., 2014*]), and become restricted to the lateral STVC following asymmetric division. However, *Htr7* mRNAs are cleared more rapidly from the FHPs, making the patterned expression easier to analyze than that of *Hand-r* (*Figures 2* and *3D*; [*Razy-Krajka et al., 2014*]). Importantly, misexpression of M-Ras^G22V using the TVC-specific *Foxf* enhancer altered cell division asymmetry and/or orientation in under 50% of the embryos, still allowing us to

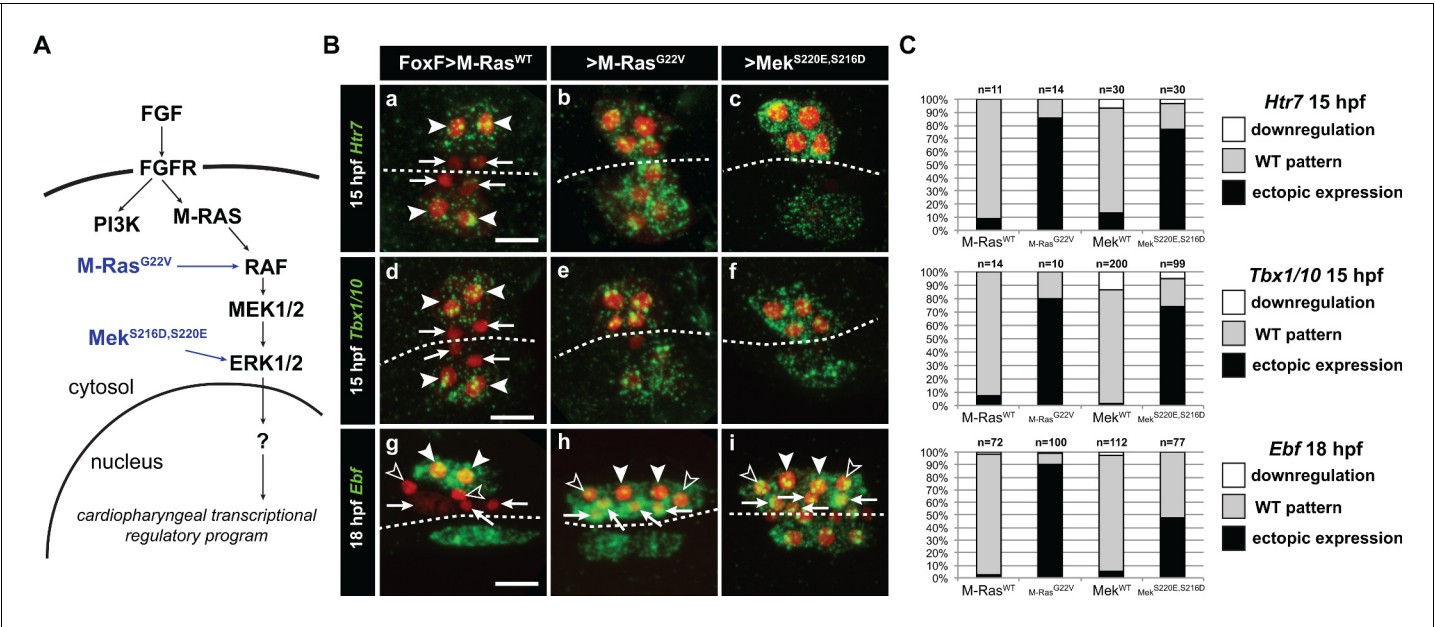

**Figure 2.** Constitutively active M-Ras and MEK are sufficient to impose a pharyngeal muscle fate in the cardiopharyngeal lineage. (**A**) Diagram of the FGF-MAPK transduction pathway with constitutive activation by M-Ras$^{G22V}$ and MEK$^{S216D,S220E}$ mutants. (**B**) Expression patterns of markers of the lateral TVC progeny, *Htr7* (a, b, c,), *Tbx1/10* (d, e, f) and *Ebf* (g, h, i), visualized by in situ hybridization following TVC-specific over-expression of M-Ras$^{WT}$ (as control), M-Ras$^{G22V}$ and MEK$^{S216D,S220E}$. M-Ras$^{WT}$ overexpression (a, d, g) does not alter the wild-type spatial expression patterns of *Htr7*, *Tbx1/10* and *Ebf* in lateral TVC derivatives (STVC and ASMF) and excluded from the median heart precursors. TVC-specific over-expression of M-Ras$^{G22V}$ (b, e, h) or MEK$^{S216D,S220E}$ (c, f, i) induces ectopic expression of STVC and/or ASMF markers (*Htr7*, *Tbx1/10* and *Ebf*) in the more median cells, that normally form cardiac precursors. Arrowheads indicate STVCs and ASMFs at 15 and 18 hpf, respectively. Arrows indicate FHPs and open arrowheads mark SHPs. At 18 hpf, the FHPs start dividing or have divided into 4 cells. Anterior to the left. Scale bar, 10 μm. (**C**) Corresponding histograms: Larvae with TVC-specific over-expression of MEK$^{WT}$ retain the wild-type expression patterns. For simplicity, ectopic expressions in half to all of the cardiac precursors were combined in the same phenotype category. 'n' corresponds to the number of embryo halves documented per condition. See also ***Figure 1—figure supplement 2***.

DOI: https://doi.org/10.7554/eLife.29656.005

The following figure supplements are available for figure 2:

**Figure supplement 1.** Effects of defined FGF-MAPK signaling perturbations on cell division patterns in the cardiopharyngeal lineage.

DOI: https://doi.org/10.7554/eLife.29656.006

**Figure supplement 2.** The constitutively active MEK$^{S216D,S220E}$ mutant is sufficient to impose a TVC identity to the whole B7.5 lineage.

DOI: https://doi.org/10.7554/eLife.29656.007

identify large lateral and small median cells in a small majority of embryos (***Figure 2—figure supplement 1***). Compared to control embryos overexpressing wild-type M-Ras (M-Ras$^{WT}$), TVC-specific gain of M-Ras function caused both persistent *Htr7* expression and ectopic activation of *Tbx1/10* in the small median cells following asymmetric divisions (***Figure 2B,C***). Similarly, *Foxf>M-Ras$^{G22V}$*-expressing 18hpf larvae displayed ectopic *Ebf* activation throughout the cardiopharyngeal mesoderm (***Figure 2B,C***). This cannot be simply accounted for by general disruption of cell division patterns at this later stage (***Figure 2—figure supplement 1***), since similar disruptions can be caused by *Foxf>dnFGFR*, which inhibits *Ebf* expression (***Figure 1C***). These results indicated that forced M-Ras activity is sufficient to upregulate STVC and ASMF markers ectopically. This is consistent with the idea that spatially defined signaling upstream of M-Ras restricts MAPK activity, thus localizing STVC- and ASM-specific gene activities.

To further probe the signal transduction pathway, we engineered a constitutively active version of the Ciona Mek1/2 protein by introducing phosphomimetic mutations of two conserved serine residues in the catalytic domain, as previously shown for the mammalian homolog (***Cowley et al., 1994***; ***Mansour et al., 1994***). Early misexpression of Mek$^{S220E,S216D}$ in the B7.5 lineage using the *Mesp* enhancer caused ectopic TVC induction, mimicking the effects of gain of Ets1/2 function (***Figure 2—figure supplement 2***; [***Davidson et al., 2006***]). As seen with M-Ras$^{G22V}$, TVC-specific misexpression

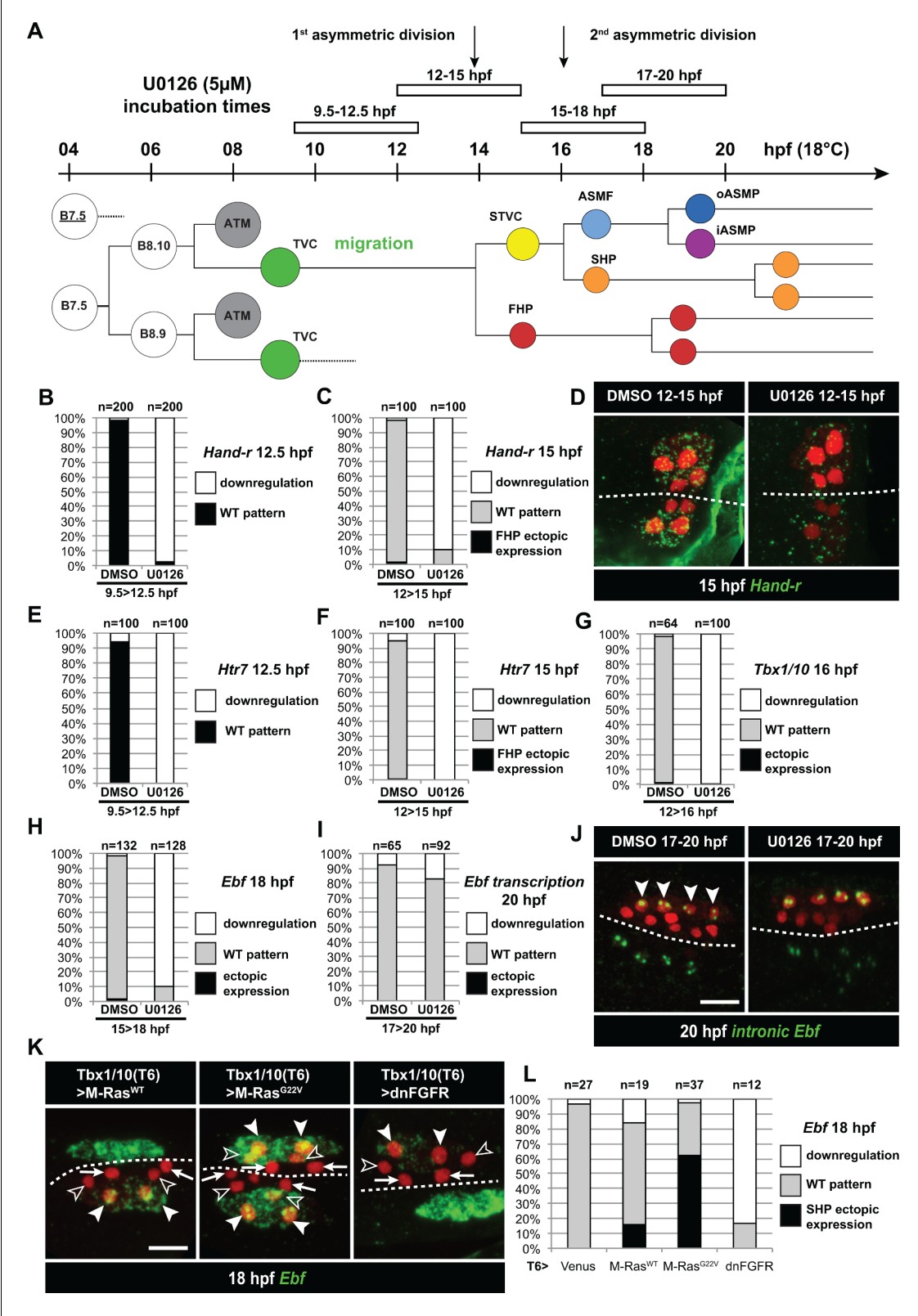

**Figure 3.** Temporal requirement for MAPK activity permits the progressive deployment of the cardiopharyngeal regulatory program. (**A**) Summary of the CPM cell lineage showing the different U0126 treatments with regard to the timing of cell divisions. Abbreviations and color codes as in *Figure 1*. (**B, C**) Proportions of embryo halves with wild-type or downregulated expression of *Hand-r* at 12.5 hpf (**B**) and 15 hpf (**C**) following 3 hr incubations in U0126 (with DMSO as control treatment). (**D**) *Hand-r* expression visualized by in situ hybridization at 15 hpf in control (DMSO treated) and U0126

*Figure 3 continued on next page*

Figure 3 continued

treated embryos. In control embryos, *Hand-r* remains expressed in the STVCs and downregulated in the FHPs. In U0126 (12–15 hpf) treated embryos, downregulation of *Hand-r* expression is observed throughout the TVC progeny (STVCs and FHPs), suggesting inhibition of transcription and inheritance of remnant transcripts following TVC divisions. (E, F) Proportions of embryo halves with wild-type or downregulated expression of *Htr7* at 12.5 hpf (E) and 15 hpf (F) following 3 hr incubations in U0126 (with DMSO as control treatment). (G) Proportions of larvae with wild-type expression or downregulated expression of *Tbx1/10* at 16 hpf following 4 hr incubation in U0126 (with DMSO as control). (H) Proportions of larvae with wild-type or downregulated expression of *Ebf* at 18 hpf following a three hour incubation in U0126 (with DMSO as control). (I) Proportions of larvae with wild-type or downregulated transcription of *Ebf* at 18 hpf following a three hour incubation in U0126 (DMSO as vehicle control). (J) Pattern of nascent *Ebf* transcripts visualized by in situ hybridization with intronic probes (green) at 20 hpf. The nuclear dots reveal the active transcription sites in the four ASMPs per side in larvae, both control/DMSO- and U0126-treated from 17 to 20 hpf. (K) *Ebf* expression (green) in 18hpf larvae expressing control M-Ras$^{WT}$, constitutively active M-Ras$^{G22V}$ or dominant negative dnFGFR under the control of the T12 element, an STVC-specific *Tbx1/10* enhancer. Arrows: first heart precursors (FHP); open arrowhead: second heart precursors (SHPs); closed arrowheads: ASM founder cells (ASMFs); dotted line: midline. (L) Proportions of larvae with wild-type or downregulated expression of *Ebf* at 18 hpf in larvae with Venus (control), M-Ras$^{WT}$, M-Ras$^{G22}$, or dnFGFR driven by *Tbx1/10* cis-regulatory sequence and overexpressed in the STVCs. 'n': number of individual halves documented per condition.

DOI: https://doi.org/10.7554/eLife.29656.008

of Mek$^{S220E,S216D}$ using the *Foxf* enhancer also caused ectopic expression of *Htr7* and *Tbx1/10*, and *Ebf* in 15 and 18hpf larvae, respectively (**Figure 2B,C**). Taken together, these results indicate that activity of the canonical FGF-Ras-MEK-ERK pathway is progressively restricted to the STVC and ASMF, and is both necessary and sufficient to promote STVC- and ASMF-specific gene expressions.

## Continuous FGF-MAPK activity is required for the successive activations of *Tbx1/10* and *Ebf*

FGF-MAPK signaling is sufficient and necessary to maintain *Hand-r* expression in late TVCs (**Figure 1**), and *Hand-r* is necessary for *Ebf* expression in the ASMF (**Razy-Krajka et al., 2014**). Therefore, it is possible that later FGF-MAPK signaling is dispensable for *Tbx1/10* and *Ebf* activation and ASM specification, as long as STVC and ASMF cells inherit sustained levels of *Hand-r* mRNAs and/or proteins. To disentangle late from early requirements of FGF-MAPK signaling, we incubated embryos at different stages with the MEK/MAPKK inhibitor U0126, which abolishes dual ERK phosphorylation and the initial MAPK-dependent TVC induction in Ciona embryos (**Figure 1—figure supplement 1**; [**Davidson et al., 2006**; **Hudson et al., 2003**]). MEK inhibition during TVC migration (i.e. between 9.5 and 12.5 hpf, **Figure 3A**) blocked the expression of *Hand-r* and *Htr7* in late TVCs (**Figure 3B,E**). U0126 treatments in late TVCs, and through the first asymmetric division (i.e. between 12 and 15 hpf, **Figure 3A**) did not alter TVC division patterns (**Figure 2—figure supplement 1**), but it blocked both the maintenance of *Hand-r* and *Htr7*, and the activation of *Tbx1/10* in the STVCs (**Figure 3C,D,F,G**). Finally, MEK inhibition in late STVCs and through asymmetric divisions (i.e. between 15 and 18 hpf) also did not alter STVC divisions (**Figure 2—figure supplement 1**), but it blocked the ASMF-specific expression of *Ebf* (**Figure 3H**). These results indicate that continuous MEK activity is required throughout cardiopharyngeal development to successively activate TVC-, STVC-, and ASMF-expressed genes.

Since *Ebf* expression is maintained for several days in the ASMF derivatives as they differentiate into body wall and siphon muscles (**Razy-Krajka et al., 2014**), we tested whether continued MEK activity is also required for the maintenance of *Ebf* expression past its onset and cells' commitment to an ASM fate. Using both regular and intron-specific antisense probes, which specifically detect nascent transcripts (**Wang et al., 2013**), we showed that later MEK inhibition (i.e. U0126 incubation between 17 and 20 hpf) did not block the maintenance of *Ebf* transcription in the ASMPs (**Figure 3I, J**). This indicates that sustained MEK activity is required until the onset of *Ebf* expression, but not beyond, and the maintenance of *Ebf* expression during ASM development is independent of MAPK.

Since U0126 treatments affect the whole embryo, we sought to further confirm the later roles for FGF-MAPK signaling specifically in the cardiopharyngeal mesoderm. To this aim, we used an STVC-specific enhancer from the *Tbx1/10* locus (termed *T6*; **Figure 3K,L**; **Figure 4—figure supplement 1**; (**Tolkin and Christiaen, 2016**; Racioppi et al., in preparation) to drive expression of either dnFGFR or the constitutively active M-Ras$^{G22V}$ starting at ~14 hpf, and assayed *Ebf* expression at 18hpf (**Figure 3K,L**). These perturbations minimally affected the cell division patterns (**Figure 2—figure supplement 1**), such that cells corresponding to FHP, SHP and ASMF could be identified by their

position relative to the midline in many embryos (*Figure 3K*). M-Ras$^{G22V}$ misexpression caused conspicuous ectopic *Ebf* expression in the SHPs, whereas dnFGFR-mediated inhibition of MAPK activity blocked *Ebf* activation in the lateral ASMFs. These results support the notion that localized FGF-MAPK activity is necessary and sufficient for ASMF-specific expression of *Ebf*.

## Coherent feed-forward circuits for cardiopharyngeal mesoderm patterning and ASM fate specification

The above results indicate that *Hand-r*, *Tbx1/10* and *Ebf* require ongoing FGF-MAPK activity for their successive activations in the TVCs, STVCs and ASMFs, respectively. We previously showed that RNAi and/or CRISPR-mediated inhibition of either *Hand-r* or *Tbx1/10* function blocks *Ebf* activation in the ASMFs, where both *Hand-r* and *Tbx1/10* expressions are maintained (*Razy-Krajka et al., 2014*; *Tolkin and Christiaen, 2016*; *Wang et al., 2013*). We used epistasis assays to systematically test whether early regulators mediate the effects of FGF-MAPK on later gene expression and ASM fate specification, or whether FGF-MAPK signaling acts both upstream and in parallel to early regulators in a more complex regulatory circuit.

We first revisited the regulatory relationships between FGF-MAPK, *Hand-r* and *Tbx1/10* in late TVCs and early STVCs. We validated single guide RNAs (sgRNAs) for CRISPR/Cas9-mediated mutagenesis of *Hand-r* (*Supplementary file 1*; [*Gandhi et al., 2017*]), and determined that *Hand-r* function is necessary for *Tbx1/10* activation in the STVCs (*Figure 4A*). Co-expression of a CRISPR-resistant *Hand-r* cDNA (Hand-r$^{PAMmis}$) rescued *Tbx1/10* expression in the STVCs, indicating that *Tbx1/10* down-regulation in this CRISPR 'background' is specifically due to *Hand-r* loss-of-function (*Figure 4A*). To further probe if *Hand-r* activity is necessary for FGF-MAPK-dependent *Tbx1/10* expression, we used gain of M-Ras function in a *Hand-r* CRISPR 'background'. Whereas, misexpression of the constitutively active M-Ras$^{G22V}$ caused ectopic *Tbx1/10* expression, concomitant loss of *Hand-r* function diminished both endogenous and ectopic *Tbx1/10* expression in the STVC and FHP, respectively (*Figure 4A*). Although, remaining ectopic activation could still be observed, possibly because M-Ras$^{G22V}$ could boost *Hand-r* expression in heterozygous cells where CRISPR/Cas9 disrupted only one copy of the gene. This data indicate that Hand-r function is necessary for FGF-MAPK-induced activation of *Tbx1/10*.

To further probe the epistatic relationships between *Hand-r* and MAPK signaling upstream of *Tbx1/10*, we attempted to rescue *Tbx1/10* expression in U0126-treated embryos, by over-expressing *Hand-r* with the TVC-specific *Foxf* enhancer. Neither did Hand-r over-expression cause ectopic *Tbx1/10* activation (in the FHPs), nor was it sufficient to rescue *Tbx1/10* expression in 15hpf STVCs (*Figure 4B*). Taken together, these data indicate that both Hand-r and MAPK activities are required to activate *Tbx1/10* in the STVCs. These results also imply that MAPK signaling is restricted to the STVC independently of Hand-r activity, which suffice to explain the STVC-specific activation of *Tbx1/10*. Finally, we isolated a minimal STVC-specific enhancer from the *Tbx1/10* locus and identified conserved putative Ets binding sites, which were necessary for reporter gene expression (*Figure 4—figure supplement 1*). This suggests that the FGF-MAPK-Ets pathway directly regulates *Tbx1/10* expression in the STVCs.

Next, we investigated the epistatic relationship between FGF-MAPK, *Hand-r*, and *Tbx1/10* upstream of *Ebf* in the ASMFs. We first used previously validated CRIPSR/Cas9 reagents targeting the *Tbx1/10* coding region (*Tolkin and Christiaen, 2016*), to confirm that B7.5-lineage-specific loss of *Tbx1/10* function inhibited *Ebf* activation, and verified that this effect could be rescued by over-expression of a CRISPR/Cas9-resistant *Tbx1/10* cDNA, expressed with a minimal TVC-specific *Foxf* enhancer (*Figure 4C*; Tbx1/10$^{PAMmis}$). In these rescue experiments, we observed ectopic *Ebf* activation in the SHP, as previously described when driving Tbx1/10 expression with a TVC-specific *Foxf* enhancer (*Wang et al., 2013*). As explained below, this ectopic activation could be attributed to a precocious expression of *Ebf* in the STVCs (*Figure 4E*). To test whether *Tbx1/10* was also required for ectopic *Ebf* expression in response to MAPK activation (see *Figure 2*), we combined CRISPR/Cas9-mediated *Tbx1/10* knockout with constitutive MAPK activation using the M-Ras$^{G22V}$ mutant and observed a significant inhibition of both endogenous and ectopic *Ebf* expression in the 18hpf ASMF and SHP, respectively (*Figure 4C*). Taken together, these results show that *Tbx1/10* function is necessary for FGF-MAPK-induced expression of *Ebf* in the ASMFs.

To further test whether *Tbx1/10* acts in parallel and/or downstream of MAPK to activate *Ebf*, we combined gain of *Tbx1/10* function with perturbations of FGF-MAPK signaling and assayed *Ebf*

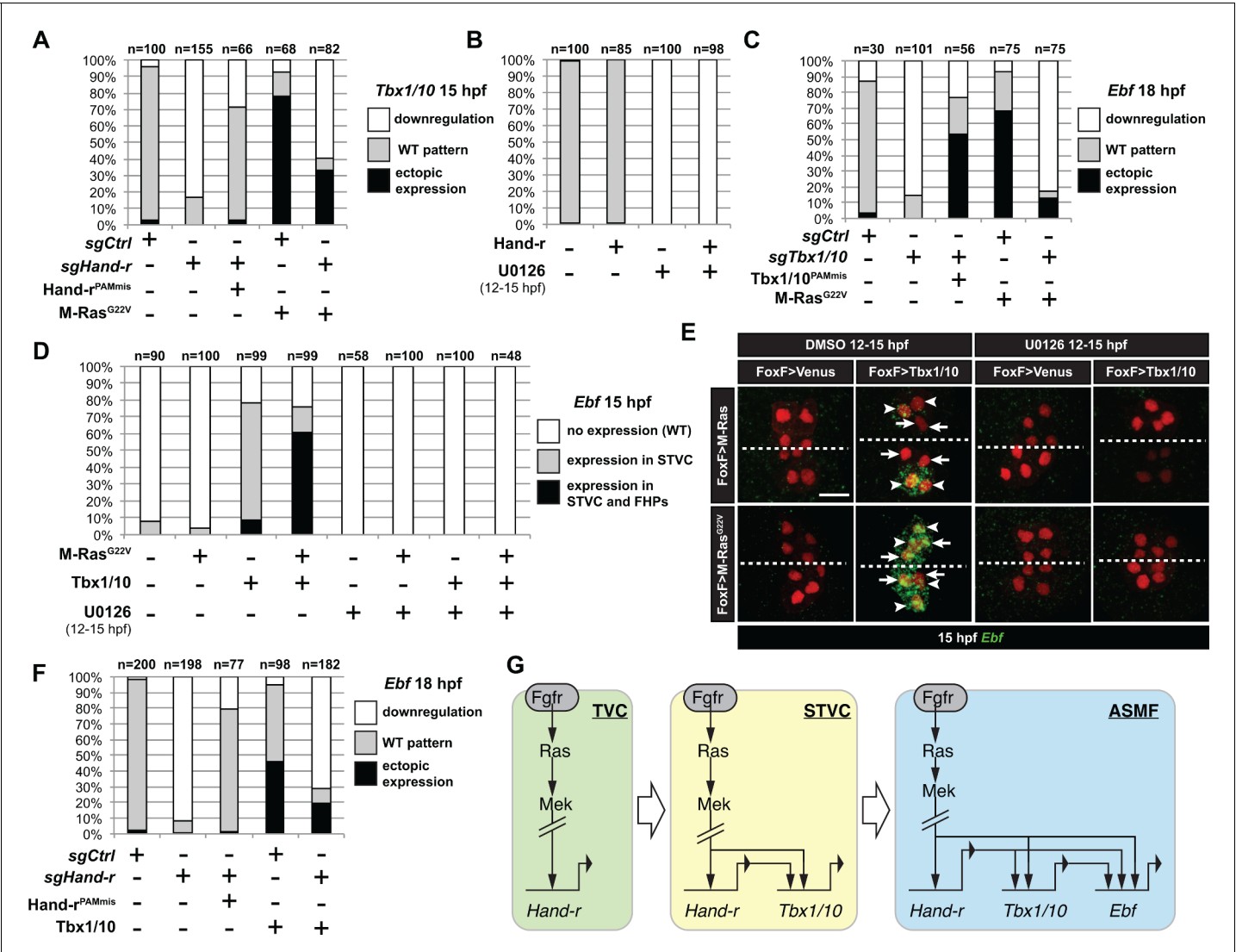

**Figure 4.** M-Ras/MAPK-driven feed-forward subcircuits control the successive activations of *Hand-r*, *Tbx1/10* and *Ebf*. (A) Proportions of embryo halves with indicated *Tbx1/10* expression patterns following TVC-specific CRISPR/Cas9-mediated mutagenesis of *Neurogenin/Neurog* as a control (*sgCtrl*), and Hand-r (*sgHand-r*). TVC-specific overexpression of a CRISPR/Cas9-resistant form of Hand-r with mutation in the PAM sequence (*Hand-r^PAMmis*) rescued *Tbx1/10* expression in the *sgHand-r* 'background'. TVC-specific overexpression of a constitutively active M-Ras mutant (M-Ras^G22) (control: M-Ras^WT) was sufficient to induce ectopic expression of *Tbx1/10* in the FHPs in *sgCtrl* embryos but not in *sgHand-r* embryos indicating that Hand-r is necessary for M-Ras-dependent activation of *Tbx1/10* transcription. (B) Proportions of embryo halves with indicated *Tbx1/10* expression patterns following TVC-specific overexpression of Hand-r or a neutral reporter (Venus) and treated from 12 to 15hpf with the MEK inhibitor U0126 (+) or with DMSO (-) as control. Hand-r overexpression is not sufficient to rescue loss of *Tbx1/10* expression due to MAPK inhibition indicating that M-Ras/MAPK activity is required in parallel of Hand-r expression to activate Tbx1/10 transcription in the TVC progeny. (C) Tbx1/10 is necessary downstream of M-Ras/MAPK activity to activate *Ebf* transcription in the TVC progeny. Shown are proportions of *Ebf* expression phenotypes following TVC-specific CRISPR/Cas9-mediated loss of *Tbx1/10* function (*sgTbx1/10*), with *Neurog*-targeting sgRNA as control (*sgCtrl*). Specificity of Tbx1/10 loss of function was validated through rescue of *Ebf* expression with TVC-specific overexpression of a CRISPR/Cas9 resistant form of *Tbx1/10* (Tbx1/10^PAMmis). Ectopic *Ebf* expression in SHPs in Tbx1/10^PAMmis larvae is explained by precocious misexpression of Tbx1/10 in the TVC as described in *Wang et al. (2013)*. TVC-specific overexpression of M-Ras^G22 (M-Ras^G22), with wild type M-Ras (M-Ras^WT) as control, was sufficient to induce ectopic expression of *Ebf* in the cardiac precursors in *sgCtrl* embryos but not in *sgTbx1/10* embryos indicating that Tbx1/10 is necessary for M-Ras-dependent activation of *Ebf* transcription. (D, E) Proportions (D) and examples (E) of 15hpf larvae halves showing indicated *Ebf* expression phenotypes in *sgCtrl* and *sgHand-r* CRISPR/Cas9 conditions combined with TVC-specific overexpression of a neutral reporter (Venus), Hand-r^PAMmis, or Tbx1/10, and with MEK inhibition by U0126 (+) or not (DMSO control (-)). Arrowhead: STVCs, Arrows: FHPs, dotted line: ventral midline (F) Loss of Hand-r function impaired the ability of Tbx1/10 to induce ectopic *Ebf* expression. For simplicity, ectopic expressions in half to all of the cardiac precursors were combined in the same phenotype category. 'n=": number of individual halves documented per condition. (G) Summary model of the temporal deployment of FGF/MAPK-driven feed-forward sub-circuits leading to the sequential activations of *Tbx1/10* and *Ebf* in the STVCs and ASMFs, respectively.

*Figure 4 continued on next page*

*Figure 4 continued*

DOI: https://doi.org/10.7554/eLife.29656.009

The following figure supplement is available for figure 4:

**Figure supplement 1.** The Tbx1/10 enhancer has conserved putative Ets binding sites required for reporter gene expression.

DOI: https://doi.org/10.7554/eLife.29656.010

expression. We realized that *Foxf*-driven misexpression of Tbx1/10 caused precocious *Ebf* activation in 15hpf STVCs (*Figure 4D,E*). This precocious expression remained remarkably patterned, suggesting that STVC-restricted FGF-MAPK activity prevented *Ebf* expression in the dpERK-negative, small median FHPs (*Figures 1B* and *4E*, *Figure 1—figure supplement 1*). Indeed, co-expression of Tbx1/10 and M-Ras$^{G22V}$ caused both precocious and ectopic *Ebf* expression in the 15hpf medial and lateral TVC derivatives, which would be FHPs and STVCs in control embryos, respectively. This data confirms that Tbx1/10 misexpression does not suffice to cause ectopic *Ebf* expression in the FHPs, because the latter presumably lack FGF-MAPK activity, as is the case in control embryos.

U0126-mediated MEK inhibition from 12 to 15hpf, that is after the onset of *Foxf>Tbx1/10* misexpression, further confirmed that MAPK activity is required in parallel to Tbx1/10 for precocious *Ebf* activation in 15hpf STVCs (*Figure 4D,E*). Taken together, these results indicate that Tbx1/10 and MAPK are both required to activate *Ebf* in the cell cycle following that of *Tbx1/10* onset.

Since *Hand-r* expression is maintained in the ASMF, and CRISPR/Cas9- or RNAi-mediated *Hand-r* knockdown blocked both *Tbx1/10* (*Figure 4A*) and *Ebf* expression (*Razy-Krajka et al., 2014*), we reasoned that Hand-r could also act both upstream and in parallel to Tbx1/10 for *Ebf* activation. To test this possibility, we assayed *Ebf* expression in 18hpf ASMF following defined perturbations of Hand-r and Tbx1/10. As expected, CRISPR/Cas9-mediated *Hand-r* mutagenesis strongly inhibited *Ebf* expression, and this effect could be rescued by a CRISPR-resistant *Hand-r* cDNA (*Figure 4F*). To test whether this effect was mediated by a loss of *Tbx1/10* expression, we attempted to rescue the Hand-r loss-of-function by over-expressing Tbx1/10 using the *Foxf* enhancer. As explained above, *Foxf*-mediated Tbx1/10 misexpression caused precocious and ectopic *Ebf* expression in larvae co-electroporated with control sgRNAs (*Figure 4D,E,F*). By contrast, combining loss of Hand-r function with Tbx1/10 misexpression inhibited both the endogenous and ectopic *Ebf* expression (*Figure 4F*), indicating that Hand-r is also required in parallel to Tbx1/10 for *Ebf* activation in the ASMFs.

Taken together, these analyses suggest that coherent feed-forward circuits govern the sequential activation of *Hand-r*, *Tbx1/10* and *Ebf* in response to continuous but progressively restricted FGF-MAPK inputs (*Figure 4G*), thus linking spatial patterning to the temporal deployment of the regulatory cascade leading to localized *Ebf* activation and pharyngeal muscle specification.

## The cell cycle entrains the temporal deployment of the cardiopharyngeal gene regulatory network

In principle, the feed-forward circuit described above is sufficient to explain the successive activations of *Hand-r*, *Tbx1/10* and *Ebf*. However, *Tbx1/10* and *Ebf* do not turn on until after TVC and STVC divisions, respectively. Notably, even when we misexpressed Tbx1/10 in the TVCs, *Ebf* was activated only after cell division and in the lateral-most cells, where FGF-MAPK signaling is normally maintained (*Figures 1B* and *4E*). This sequence of events -divisions followed by gene activation- is paramount as it permits the birth of first and second heart precursors, whose fates are antagonized by Tbx1/10 and Ebf (*Razy-Krajka et al., 2014*; *Stolfi et al., 2010*; *Wang et al., 2013*). Therefore, we sought to investigate the role(s) of the cell cycle in controlling the timing of *Tbx1/10* and *Ebf* activation.

We first evaluated the effects of cytochalasin B, a classic inhibitor of cytokinesis widely used to study cell fate specification in ascidians (*Figure 5A*; [*Whittaker, 1973*]). Treatments starting before TVC divisions (12 hpf) did not block *Tbx1/10* or *Ebf* expression in embryos fixed after their normal onset at either 16 or 19hpf, respectively (*Figure 5B,C*). Similarly, treatment starting between the first and second asymmetric divisions (15hpf) did not block localized *Ebf* expression at 19hpf (*Figure 5C*). This indicates that *Tbx1/10* and *Ebf* activations occur by default in the absence of cytokinesis, most likely because FGF-MAPK signaling persists throughout the shared cytoplasm. This data thus

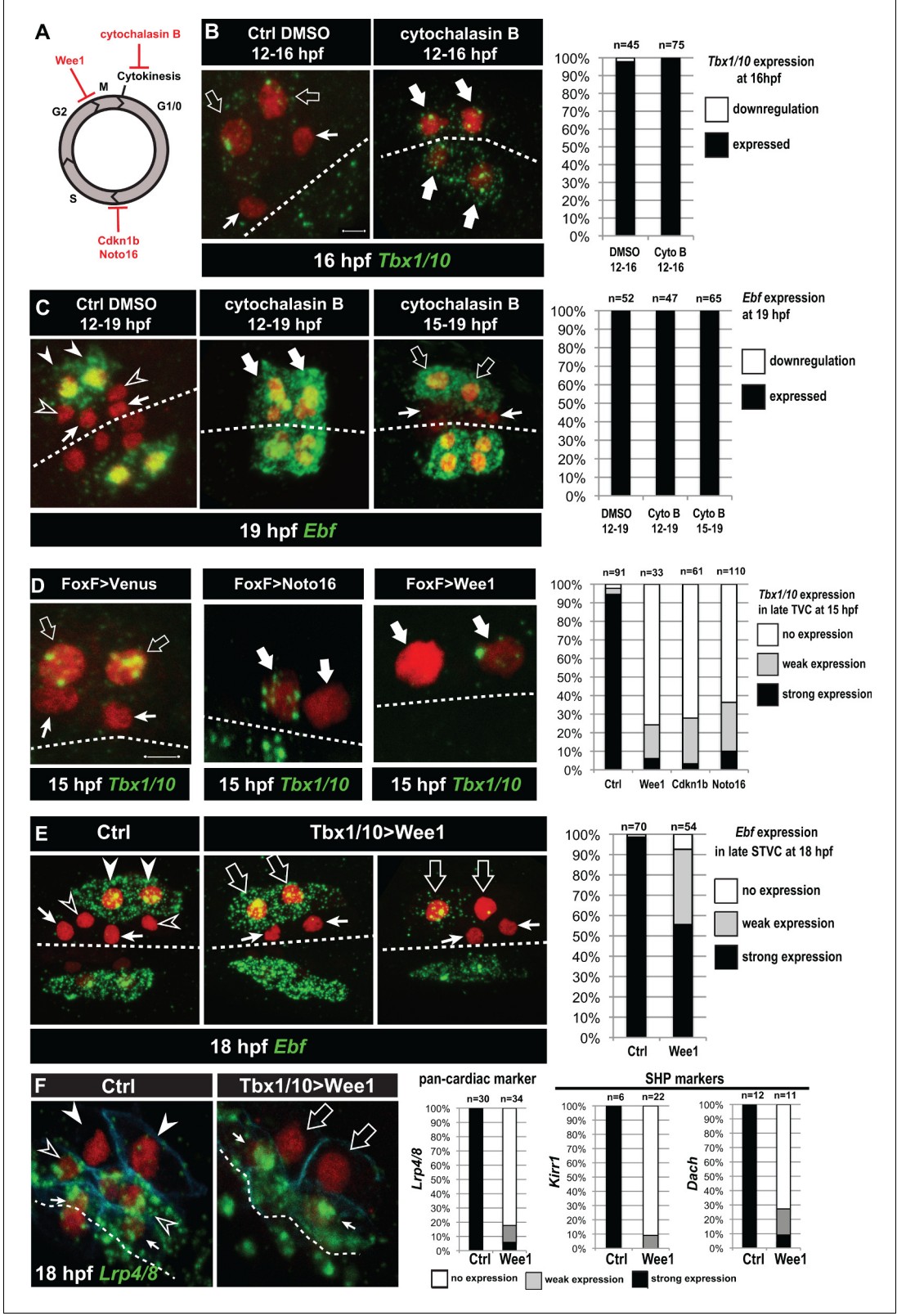

**Figure 5.** Temporal deployment of the cardiopharyngeal network is partially coupled with cell cycle progression. (A) Schematic representation of the canonical eukaryotic cell cycle, and actions of the perturbations used in this study. (B,C) Tbx1/10 and Ebf expression at indicated time points, and following inhibition of cytokinesis by cytochalasin B treatment at indicated time points. Note that 15 to 19hpf treatment is applied AFTER the first division and birth the FHPs, which do not activate *Ebf* at 19hpf (right panel, arrows). (D) Inhibition of G1/S or G2/M blocks TVC division, and reduces

*Figure 5 continued on next page*

*Figure 5 continued*

*Tbx1/10* expression. Pictures shows TVCs that have divided in controls but not in experimental cells, with one cell occasionally turning on *Tbx1/10*, but not the other. Left: the proportions of embryos showing strong *Tbx1/10* expression is substantially reduced compared to control embryos (e.g. *Figure 1*, and [*Wang et al., 2013*]). (E) Inhibition of G2/M in the STVCs by misexpression of Wee1 using the *Tbx1/10 T6* enhancer inhibits STVC division, and has a mild impact on *Ebf* expression at 18hpf. Open arrows indicate STVCs that have not divided, but express high (middle) or low (right) levels of *Ebf*. Left: control larva showing high *Ebf* expression in the ASMF (closed arrowheads), but neither in the SHPs (open arrowheads) nor in the FHPs (Arrows). (F) Misexpression of Wee1 using the *Tbx1/10 T6* enhancer (*Tbx1 >Wee1*) inhibits STVC division. Right: the proportions of embryos showing strong *Lrp4/8, Kirr1* or *Dach* expression into late STVCs is reduced compared to SHPs in control embryos (T6 >NLS::LacZ) at 18 hpf. Notably, the pan-cardiac marker *Lrp4/8* is still expressed in FHPs (arrows). Nuclei are marked in red with Mesp >NLS::lacZ, membranes in blue with Mesp >hCD4::mCherry, FHP labeled with arrows and SHP with arrowheads. 'n=', number of individual halves scored per condition. Scale bar, 5 μm. In all image panels, dotted line: ventral midline.

DOI: https://doi.org/10.7554/eLife.29656.011

The following figure supplement is available for figure 5:

**Figure supplement 1.** Dynamics of Ebf upregulation and entrainment by the cell cycle.

DOI: https://doi.org/10.7554/eLife.29656.012

illustrates how the spatial restriction of FGF-MAPK signaling, following cell divisions, leads to the localized activations of *Tbx1/10* and *Ebf*.

Cytochalasin treatments usually lead to the formation of polynucleated cells (e.g. *Figure 5C*, middle panel), because the cell cycle and nucleokinesis continue. To alter cell cycle progression more comprehensively, and specifically in the cardiopharyngeal lineage, we used genetically encoded inhibitors of cell cycle transitions: Cdkn1b.a and Cdkn1b.b (also known as Noto16), the ortholog of which is a potent inhibitor of the G1/S transition in the ascidian species *Halocynthia roretzi* (*Kuwajima et al., 2014*), and the G2/M inhibitor Wee1 (*Dumollard et al., 2017*). We used the TVC-specific *Foxf* enhancer to misexpress these negative regulators of cell cycle progression, monitored cell divisions and assayed *Tbx1/10* expression at 15hpf, when control TVCs have divided and the lateral-most STVCs normally express *Tbx1/10*. Each perturbation efficiently inhibited TVC divisions, such that only two cells were visible on either side of the embryos (*Figure 5D*). In these delayed TVCs, *Tbx1/10* expression was strongly reduced compared to control STVCs (*Figure 5D*). However, 20% to 40% of the delayed TVCs expressed *Tbx1/10* to variable extents. This suggests that the cardiopharyngeal regulatory network can qualitatively unfold independently of cell cycle progression, but the latter is necessary for *Tbx1/10* expression to its wild-type levels.

We next used the STVC-specific *Tbx1/10 T6* enhancer (*Figure 4—figure supplement 1*), to misexpress Cdkn1b.a, Noto16 and Wee1, and assayed *Ebf* expression at later stages. Inhibitors of the G1/S transition failed to block STVC divisions (data not shown), most likely because *Tbx1/10(T6)*-driven products did not accumulate quickly enough to interfere with the G1/S transition in STVCs, since this cell cycle lasts only ~2 hr compared to ~6 hr for the TVC interphase. Therefore, we focused the analyses of *Ebf* response to cell cycle perturbations on misexpression of the G2/M inhibitor Wee1. Analyses of 18hpf larvae, fixed approximately 2 hr after the documented onset of *Ebf* expression in ASMFs (*Razy-Krajka et al., 2014*), indicated that *Ebf* can turn on in arrested STVCs that failed to divide upon Wee1 misexpression (*Figure 5E*).

Because ~30% of the embryos showed variable expression, as was the case for *Tbx1/10* in the previous experiment, we reasoned that perturbations of the G2/M transition could alter the dynamics of *Ebf* upregulation. We investigated this possibility using embryos fixed every 30 min between 15.5hpf and 18hpf, when cells transition from a late *Tbx1/10*+; *Ebf*- STVC state to a committed *Ebf*+, *Mrf*+ ASMF state (*Razy-Krajka et al., 2014*; *Wang et al., 2013*). First, we scored the proportions of embryos with delayed STVCs or conspicuous ASMFs at each time point and showed that Wee1 misexpression strongly delays cell cycle progression, blocking cell divisions in a substantial fraction of embryos (*Figure 5—figure supplement 1*).

The proportion of *Ebf*+ ASMFs in control embryos progressively increased from ~20% at 15.5hpf to >90% by 18hpf, revealing an under-appreciated dynamic at the onset of *Ebf* expression, which appears to take at least one hour to be 'strongly' expressed in >75% of newborn ASMFs (*Figure 5—figure supplement 1*).

To evaluate the impact of Wee1-induced mitosis inhibition on *Ebf* accumulation, we focused on undivided STVCs at each time point (hence the lower numbers in *Figure 5—figure supplement 1A*

compare to *Figure 5—figure supplement 1B*). By 17hpf, Wee1-expressing delayed STVCs showed 'strong' *Ebf* expression in proportions as high as for control ASMFs. However, these proportions were significantly lower at 16 and 16.5hpf (Chi-square tests, p=0.002 and p=0.0003, respectively), with ~1.5 and ~1.2 times less 'strongly' expressing cells than in the control distributions (hypergeometric tests, p=0.0005 and p=0.0001, respectively). These semi-quantitative analyses suggest that the cardiopharyngeal network can eventually unfold independently of cell divisions, leading to high levels of *Ebf* expression, albeit with a delay. This suggests that STVC division entrains *Ebf* upregulation in early ASMFs.

Finally, we reasoned that Wee1-expressed delayed STVC that activate *Ebf* would not turn on heart markers (*Wang et al., 2017*). Indeed, delayed STVCs failed to activate the pan-cardiac *Lrp4/8*, and the SHP-markers *Kirr* and *Dach* (*Figure 5F*; [*Wang et al., 2017*]). This indicated that coupling STVC division with localized FGF-MAPK activity and timed *Ebf* upregulation permits the localized activation of *Ebf* and the emergence of cardiac progenitors.

## Transition from a MAPK-dependent to a MAPK-independent and autoregulative mode of *Ebf* expression in early ASMFs

We sought to further probe the mechanisms that regulate the initiation of *Ebf* expression in early ASMFs, and their biological significance for fate specification. Since we observed a progressive accumulation of *Ebf* mRNAs, and a transition from a MAPK-dependent onset to a MAPK-independent maintenance of *Ebf* transcription (*Figure 3I,J*), we reasoned that the window of MAPK-dependence might coincide with the accumulation of *Ebf* mRNAs between 16 and 17hpf. To test this possibility, we treated embryos with the MEK inhibitor U0126 at successive time points, assayed ongoing transcription using intronic probes and counted the numbers of *Ebf* transcribing cells (*Figure 6A*). This analysis revealed that *Ebf* transcription gradually lost its sensitivity to MAPK inhibition between 16 and 17hpf, that is during the first hour of the ASMF cycle when *Ebf* mRNAs normally accumulate (*Figure 5—figure supplement 1A,B*).

Because *Ebf* transcription becomes independent from MAPK by the time *Ebf* mRNA have accumulated to 'high' levels, and because *Ebf* expression lasts for several days in the progeny of the ASMFs, we reasoned that autoregulation might suffice to maintain high levels of *Ebf* mRNA past the MAPK-dependent onset. To test this possibility, we misexpressed the Ebf coding sequence using the STVC-specific *T6* enhancer as described (*Tolkin and Christiaen, 2016*). Assaying endogenous *Ebf* transcription with intronic probes demonstrated that, in addition to its normal expression in the ASMFs, Ebf misexpression caused precocious and ectopic activation of the endogenous locus in the STVCs, and in the MAPK-negative SHPs, respectively (*Figure 6C–F*). This result suggests that *Ebf* transcription bypasses both requirements for cell-division coupling and MAPK inputs if high levels of *Ebf* gene products are present in the cell.

We reasoned that, if high levels of *Ebf* expression can promote its own transcription independently of MAPK signaling, then Ebf misexpression should be sufficient to rescue a chemical inhibition of MAPK at a critical stage. We tested this possibility by combining Ebf misexpression using the STVC-specific *T6* enhancer and U0126 treatments starting at 16hpf, which normally block *Ebf* expression (*Figure 6A,D–F*). We observed that transcription of the endogenous *Ebf* locus became independent of early MAPK activity upon misexpression of an Ebf cDNA, further supporting the notion that high levels of *Ebf* expression suffice to maintain *Ebf* transcription independently of MAPK activity.

A potentially important implication of this transient MAPK requirement is to render *Ebf* expression initially reversible. For instance, *Ebf* occasionally turns on precociously in the STVCs of a small proportion of embryos (*Figure 5—figure supplement 1C*). Given the powerful anti-cardiogenic effects of Ebf (*Razy-Krajka et al., 2014*; *Stolfi et al., 2010*), persistent *Ebf* expression would have dramatic consequences for SHP development (*Wang et al., 2013*). However, because MAPK activity is excluded from the SHPs, and the early phase of *Ebf* expression depends upon continuous MAPK activity, we surmise that *Ebf* cannot be maintained in the SHPs. For instance, when embryos from the same electroporated batch were fixed at the time of early U0126 treatment (i.e. 15.75 and 16.25hpf) and ~4 hr later, at 20hpf, and assayed for *Ebf* transcription using intronic probes, initially wild-type patterns of *Ebf* transcription could not be maintained (*Figure 6—figure supplement 1A*). This suggests that, although *Ebf* can be activated precociously in a MAPK-dependent manner, its expression shuts off in the SHPs upon MAPK inhibition following STVC division.

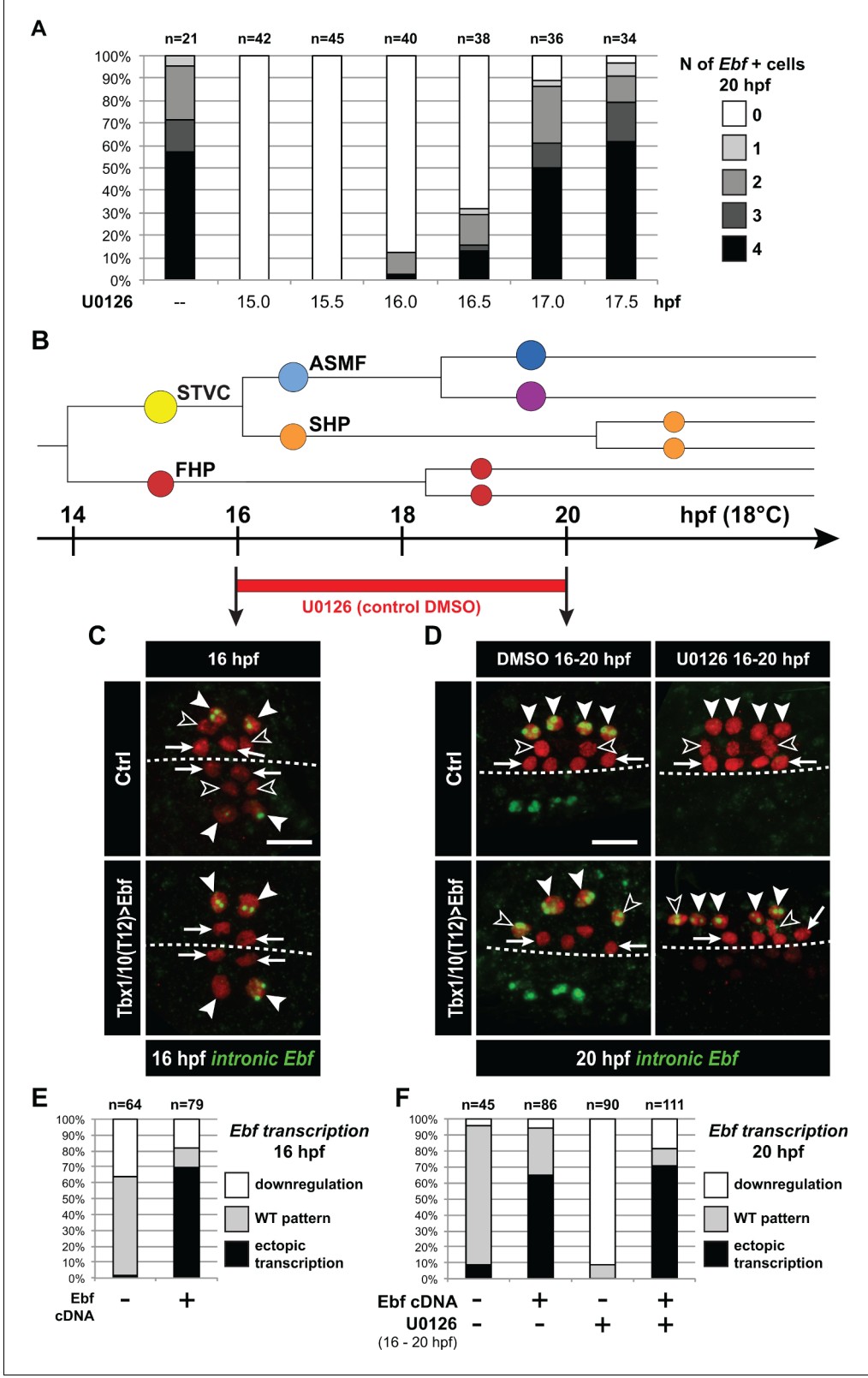

**Figure 6.** *Ebf* regulation transitions from MAPK-dependent to autoregulative during the early phase of ASMF cycle. (A) Proportions of 20hpf larva halves showing the indicated number of Ebf-expressing cells following U0126 treatments started at the indicated time points. This indicates that, by 17hpf, Ebf expression, which started at ~16 hpf, has become largely insensitive to loss of MAPK activity. (B) Summary lineage diagram and time scale indicating the approximate stages for U0126 and DMSO (control) treatments for the results shown in (C, D). (C) Control (Ctrl) and Ebf-misexpressing
*Figure 6 continued on next page*

*Figure 6 continued*

embryos fixed at 16hpf, prior to chemical treatments, and stained for nascent transcripts with an intronic *Ebf* probe. In controls, the ASMFs (solid arrowhead), but neither the SHPs (open arrowheads) nor the FHPs (arrows), actively transcribe *Ebf* (green nuclear dots). In Larvae misexpressing the Ebf cDNA under the control of the STVC-specific Tbx1/10 enhancer, divisions are delayed and STVCs (solid arrowheads) activated transcription of endogenous *Ebf* loci (green nuclear dots). (D) After 4 hr, U0126 treated ASMFs no longer transcribe Ebf (top right image, solid arrowheads), whereas control DMSO-treated ASMFs do (top left, green nuclear dots). Upon misexpression of the Ebf cDNA in the STVCs and derivatives, ongoing *Ebf* transcription is detected at 20hpf in both DMSO and U0126-treated cells, and it persists in both ASMFs (solid arrowheads), and SHPs (open arrowheads). (E, F) Proportions of larvae halves showing the indicated *Ebf* transcription patterns, in indicated experimental conditions, as illustrated in C and D, respectively.

DOI: https://doi.org/10.7554/eLife.29656.013

The following figure supplement is available for figure 6:

**Figure supplement 1.** MAPK signaling is necessary for *Ebf* expression only in early ASMF, and cell cycle inputs shorten the MAPK-dependent period.
DOI: https://doi.org/10.7554/eLife.29656.014

We further addressed the interplay between cell division, MAPK signaling and *Ebf* expression. We reasoned that, if cell divisions entrain Ebf accumulation and the transition to a MAPK-independent autoregulative mode, then delaying STVC divisions should extend the period of MAPK-dependent *Ebf* transcription. We tested this possibility by expressing Wee1 under the control of the STVC-specific *T6* enhancer, and treated embryos with U0126 at 17hpf, which inhibited the maintenance of *Ebf* transcription in only 15% to 20% of the control embryos (*Figure 6A*, *Figure 6—figure supplement 1B*). The proportion of embryos showing U0126-sensitive *Ebf* transcription increased to almost 50% upon *Tbx1/10>Wee1* expression (*Figure 6—figure supplement 1B*), which is consistent with our hypothesis that inhibiting the G2/M transition delays the accumulation of *Ebf* gene products, thus postponing the transition from a low level/MAPK-dependent to an high level/MAPK-independent and self-activating mode of *Ebf* regulation.

We propose a model for *Ebf* regulation whereby Hand-r, Tbx1/10, ongoing MAPK signaling and cell-cycle-regulated transcriptional input(s) govern the onset and initial accumulation of *Ebf* gene products during the first hour of the ASMF cycle, whereas the maintenance of *Ebf* expression relies primarily on MAPK-independent autoactivation, following initial accumulation (*Figure 7*).

## Discussion

Here, we demonstrated that the progressive restriction of FGF-MAPK signaling follows oriented and asymmetric cell divisions of multipotent progenitors and patterns the ascidian cardiopharyngeal mesoderm in space and time. Dynamic FGF-MAPK activity patterns lead to the localized expression of *Hand-r*, *Tbx1/10* and *Ebf* in fate-restricted pharyngeal muscle precursors, and their concomitant exclusion for first and second heart precursors. We show that coherent feedforward circuits encode the successive activations of *Hand-r*, *Tbx1/10* and *Ebf*, whereas cell divisions entrain the progression of this regulatory sequence and thus define the timing of gene expression. Finally, we provide evidence that the initiation of *Ebf* expression depends on MAPK activity in early ASMF, until Ebf accumulation permits MAPK-independent auto-activation. Given the potent anti-cardiogenic, and pro-pharyngeal muscle effects of Ebf (*Razy-Krajka et al., 2014*; *Stolfi et al., 2010*), we surmise that the latter switch corresponds to the transition from a cardiopharyngeal multipotent state to a committed pharyngeal muscle identity.

### Spatial patterning by localized maintenance of FGF-MAPK signaling

Our results demonstrate that MAPK signaling is maintained only in the lateral-most daughter cells following each asymmetric division of multipotent cardiopharyngeal progenitors - the TVCs and STVCs. This asymmetric maintenance is necessary and sufficient for the progressive and localized deployment of the pharyngeal muscle network. Notably, the TVCs themselves are initially induced by similar polarized FGF-MAPK signaling coincidental to asymmetric cell divisions of their mother cells, the B8.9 and B8.10 founder cells (*Davidson et al., 2006*). At this stage, asymmetrical maintenance of sustained FGF-MAPK signaling involves intrinsic Cdc42-dependent polarity of the founder cells, which promotes polarized cell-matrix adhesion of the prospective TVC membrane to the ventral epidermis. The latter differential integrin-mediated adhesion promotes localized MAPK

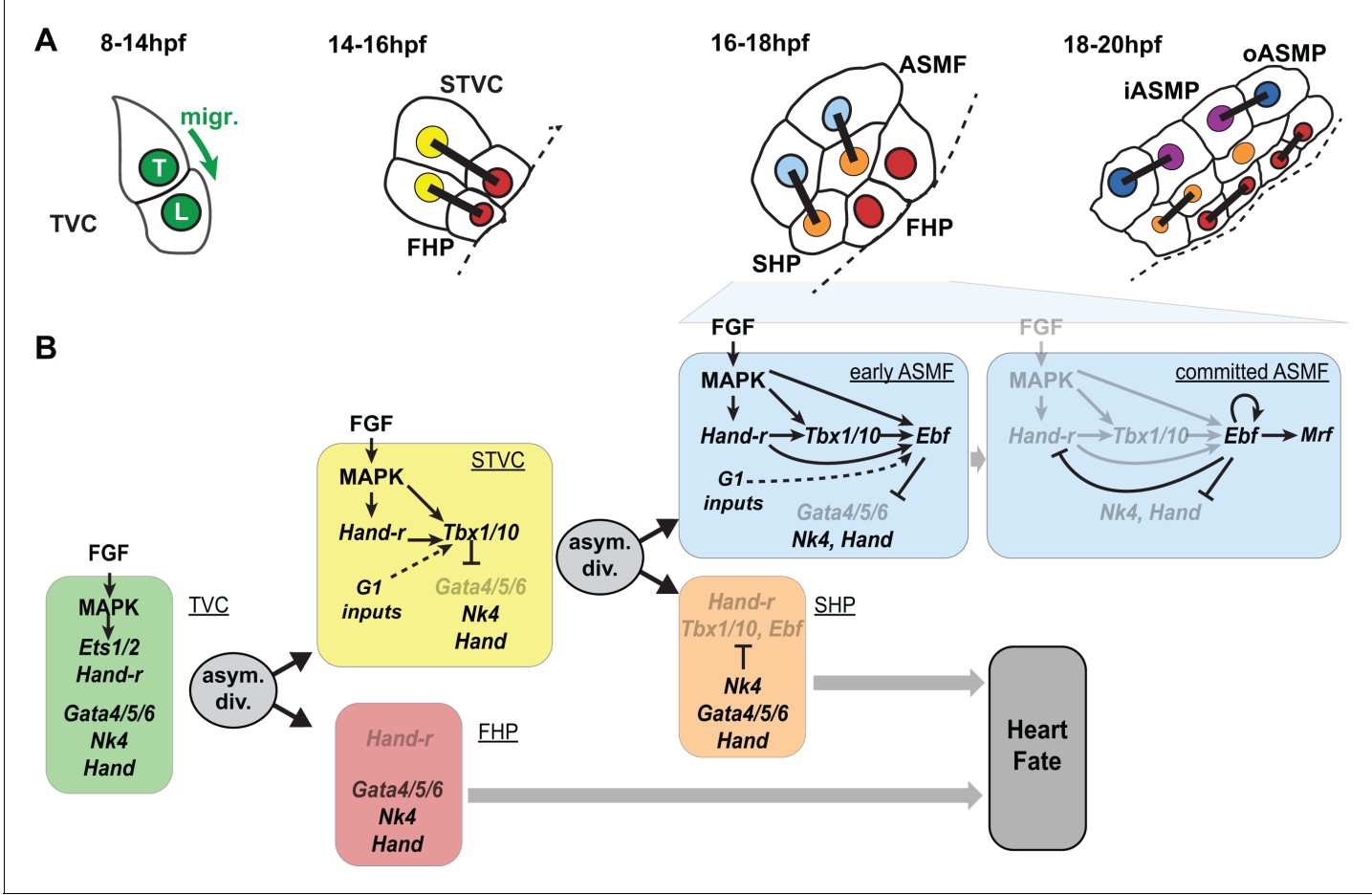

**Figure 7.** Summary model. (**A**) Schematic representation of cardiopharyngeal lineage cells at successive time points representing the main fate transitions. hpf: hours post-fertilization; TVC: trunk ventral cells; L: Leader T: trailer; migr.: migration; STVC: second trunk ventral cells; FHP: first heart precursors; dotted line: midline; black bars link sister cells; ASMF: atrial siphon muscle founder cells; SHP: second heart precursors; iASMP: inner atrial siphon muscle precursors; oASMP: outer atrial siphon muscle precursor (these cells correspond to stem-cell-like *Mrf*-; Notch+ precursors and *Mrf*+; Notch- differentiating myoblasts, respectively; see (*Razy-Krajka et al., 2014*) for details). (**B**) Lineage diagram and documented regulatory relationships between indicated genes and pathways, as showing here and in (*Razy-Krajka et al., 2014*; *Wang et al., 2013*). In TVCs, primed heart and ASM markers are coexpressed, and maintenance of the STVC and ASM markers requires ongoing FGF/MAPK signaling. Following the first oriented and asymmetric cell division, FGF-MAPK is maintained only in the STVCs, which permits the continued expression of Hand-r and the activation of *Tbx1/10*. Cell division, presumably through G1-specific inputs, contributes to *Tbx1/10* activation, and Tbx1/10 function antagonizes *Gata4/5/6* expression (*Wang et al., 2013*). In the FHPs, termination of FGF-MAPK signaling inhibits *Hand-r* expression and prevents *Tbx1/10* activation. Following oriented and asymmetric division of the STVCs, FGF/MAPK signaling persists only in the ASMFs, where it permits the transient maintenance of *Hand-r* and *Tbx1/10*, both of which act in parallel to FGF/MAPK to activate *Ebf* expression, together with contributions from presumed G1 inputs. Ebf activities further antagonize the cardiac program (marked by *Gata4/5/6*, *Nk4/Nkx2.5* and *Hand* expression; [*Razy-Krajka et al., 2014*; *Stolfi et al., 2010*; *Wang et al., 2013*]). Once Ebf expression reaches 'high levels', its regulation becomes MAPK-independent and self-activating (this study). It also feeds back negatively on early activators such as Hand-r, and promotes the expression of the muscle determinant *Mrf* (*Razy-Krajka et al., 2014*; *Tolkin and Christiaen, 2016*). We propose that this transition represents commitment to an ASM fate. In the SHPs, termination of FGF/MAPK signaling prevents maintenance of Hand-r and Tbx1/10 expression, which, together with repressive inputs from Nk4/Nkx2.5, inhibits *Ebf* activation (*Wang et al., 2013*), and permits heart fate specification (*Wang et al., 2017*).

DOI: https://doi.org/10.7554/eLife.29656.015

activation, leading to TVC induction (*Cooley et al., 2011*; *Norton et al., 2013*). It has been proposed that adhesion- and caveolin-dependent polarized FGFR recycling during mitosis accounts for the localized activation of MAPK in the prospective TVCs (*Cota and Davidson, 2015*). Whereas similar mechanisms could in principle account for asymmetric maintenance of FGF-MAPK signaling in STVCs and ASMFs, this has not been formally tested and there are notable differences opening the possibility that other mechanisms may be at work: during TVC induction, MAPK signaling is

maintained in the smaller daughter cell that contacts the epidermis, whereas in the following divisions, MAPK activity persists in the larger daughter cells and all cells maintain contact with the epidermis (Kaplan et al., in preparation). Moreover, using fusion proteins as in previous studies, we could not observed a marked polarized distribution of FGFR molecules to the lateral-most cells (the STVCs and ASMFs; Kaplan et al., in preparation). However, the fact that constitutively active forms of M-Ras and Mek1/2 were sufficient to bypass the loss of MAPK activity, and impose pharyngeal muscle specification, indicates that differential FGF-MAPK activity is regulated upstream of M-Ras. Further work is needed to elucidate the cellular and molecular mechanisms governing the spatiotemporal patterns of FGF-MAPK signaling in the cardiopharyngeal mesoderm. In particular, it will be important to disentangle the relative impacts of extrinsic (i.e. tissues, contacts) vs. intrinsic (i.e. asymmetric cell division) effects onto FGF-MAPK signaling and the downstream transcriptional inputs.

Our preliminary analyses indicate that perturbations of the FGF-Ras-MEK pathway can alter cardiopharyngeal cell division patterns (*Figure 2—figure supplement 1*). While these effects did not account for the observed changes in gene expression, future studies will unravel FGF-MAPK-dependent and cardiopharyngeal-specific mechanisms governing the orientation and asymmetry of progenitor cell division (Kaplan et al., in preparation).

## Transcriptional consequences of differential FGF-MAPK signaling

Differential FGF-MAPK signaling rapidly impacts cell-specific gene expression, we thus surmise that transcriptional effectors are dynamically regulated. Even though we have not formally identified the downstream DNA-binding transcription factor (see discussion below), it is conceivable that the phosphorylated forms of either transcriptional effector could persist through cell division upon maintenance of FGF-MAPK activity. However, we have shown that continuous MAPK activity is needed following each division. Therefore, we must invoke elusive phosphatase activities, such as dual-specificity phosphatases (DUSPs; [*Patterson et al., 2009*]), which would reset transcriptional effectors to a dephosphorylated state, thus rendering steady-state FGF-Ras-MAPK inputs necessary.

Systematic dephosphorylation of FGF-MAPK transcriptional effectors is particularly important for heart fate specification. As shown in our companion paper (*Wang et al., 2017*), whole genome analyses indicate that heart-specific de novo gene expression requires MAPK inhibition (*Wang et al., 2017*). The molecular mechanisms remain elusive, but one simple possibility is that, lest fate-restricted heart precursors inhibit MAPK activity, they will activate *Tbx1/10* and *Ebf*, which will block the cardiac program (*Razy-Krajka et al., 2014*; *Stolfi et al., 2010*; *Wang et al., 2013*).

Finally, we previously proposed that repressor inputs from Nk4/Nkx2-5 are needed in the second heart precursors to avoid ectopic activation of *Ebf* (*Wang et al., 2013*). The observation that *Nk4* transcripts are detected in all cardiopharyngeal cells opened the question as to how *Ebf* would escape repression by Nk4 in the ASMFs. Differential MAPK activity offers an intriguing possibility: for instance, Nk4/Nkx2.5-mediated repression in other species involves the co-repressor Groucho/TLE (*Choi et al., 1999*), which is strongly expressed in the cardiopharyngeal mesoderm (*Razy-Krajka et al., 2014*); and, in flies, MAPK-mediated phosphorylation of Groucho inhibits its repressor function (*Cinnamon et al., 2008*; *Cinnamon and Paroush, 2008*; *Hasson et al., 2005*). Therefore, it is possible that persistent MAPK signaling dampens Groucho/TLE-mediated repressive inputs on cell-specific regulatory genes like *Ebf*. Future studies will determine whether such mechanisms provide bistable switches underlying MAPK-dependent fate choices in the cardiopharyngeal mesoderm.

## Temporal deployment of the pharyngeal muscle network

The localized and successive activation of *Tbx1/10* and *Ebf* in STVCs, and ASMFs, respectively, are important features of the cardiopharyngeal network that permit the emergence of diverse cell fates: first and second heart precursors, and atrial siphon muscle precursors. Experimental misexpression of Ebf throughout the cardiopharyngeal mesoderm suffice to inhibit heart development (*Razy-Krajka et al., 2014*; *Stolfi et al., 2010*), illustrating the importance of *Ebf* restriction to the ASMF for the emergence of first and second heart precursors.

Our analyses indicate that the sequential activations of *Hand-r*, *Tbx1/10* and *Ebf* is encoded in the feed-forward structure of this sub-circuit, whereas the continuous requirement for MAPK inputs and their progressive exclusion from heart progenitors restrict the competence to activate *Tbx1/10* and *Ebf* to the most lateral cells, after each division. Our model implies that each gene may directly

respond to transcriptional inputs from MAPK signaling. We have not formally identified the transcription factors(s) that mediate the transcriptional response to FGF-MAPK signaling. However, multipotent cardiopharyngeal progenitors express *Ets1/2* and *Elk*, two common transcriptional effectors of FGF/MAPK signaling in *Ciona* (*Bertrand et al., 2003*; *Christiaen et al., 2008*; *Davidson et al., 2006*; *Gainous et al., 2015*), Ets1/2 has been implicated in the initial FGF-MAPK-dependent induction of multipotent TVCs (*Christiaen et al., 2008*; *Davidson et al., 2006*), and its expression is also progressively restricted to the lateral-most progenitors following each division (*Razy-Krajka et al., 2014*). Taken together, Ets1/2 and, to some extend, Elk are intriguing candidate transcriptional effectors of FGF/MAPK signaling in cardiopharyngeal development.

The binding preferences of Ets-family factors have been extensively studied in *Ciona*, and they do not depart markedly from conserved Ets-family binding sites with a GGAW core (*Bertrand et al., 2003*; *Farley et al., 2015, 2016*; *Gueroult-Bellone et al., 2017*; *Khoueiry et al., 2010*). Putative Ets-family binding sites in the TVC-specific *Hand-r* enhancer are conserved between *Ciona intestinalis* and its sibling species *C. robusta* and *C. savignyi*, and necessary for its activity in reporter assays (*Woznica et al., 2012*). Similarly, minimal STVC and ASM enhancers for *Tbx1/10* and *Ebf*, respectively, contain conserved putative Ets-family binding sites, which contribute to proper reporter gene expression in transfection assays (*Figure 4—figure supplement 1*; (*Razy-Krajka et al., 2014*; *Wang et al., 2013*) and data not shown). Taken together, these observations suggest that the proposed feed-forward sub-circuit involves direct transcriptional inputs from FGF-MAPK-regulated Ets-family factors on the cardiopharyngeal enhancers of *Hand-r*, *Tbx1/10* and *Ebf*.

Whereas the regulatory architecture of the MAPK; Hand-r; Tbx1/10; Ebf sub-circuit explains the sequence of activation events, it is also crucial for its correct deployment, and the generation of diverse cell identities, that genes are not fully activated before successive cell divisions. While divisions are not absolutely required for *Ebf* to eventually turn on, cell cycle progression appears to entrain the deployment of this network, especially for *Tbx1/10* and *Ebf* activation in STVCs and ASMFs, respectively. These observations imply that the intrinsic dynamic of the network is slower than observed. This allows first and second heart precursors to be born prior to the onset of *Tbx1/10* and *Ebf*, respectively. The latter sequence is essential for the heart progenitors to escape the anti-cardiogenic effects of Tbx1/10 (*Wang et al., 2013*), and Ebf (*Razy-Krajka et al., 2014*).

Initial *Ebf* expression in early ASMFs is also labile and MAPK-dependent for approximately one hour. This continued requirement for MAPK inputs ensures that, in the rare instances when *Ebf* expression starts in the multipotent STVC progenitors and/or expands to the nascent SHPs, inhibition of MAPK shuts off *Ebf* expression before it reaches the levels needed for commitment to an ASM fate. Indeed, our results indicate that, once *Ebf* mRNAs have accumulated to high levels, its expression becomes auto-regulative and MAPK-independent. We surmise that this transition coincides with a fundamental switch from a multipotent cardiopharyngeal state to a committed pharyngeal muscle identity.

From this standpoint, the observed entrainment of *Ebf* expression by the cell cycle can be seen as acceleration of the transition to commitment following asymmetric division of multipotent progenitors. Although the mechanisms remain elusive, it is likely that this requires the M/G1 transition, as the G1 phase has been shown to be particularly conducive to the expression of fate-specific regulators in mammalian pluripotent stem cells (*Dalton, 2015*; *Pauklin et al., 2016*; *Pauklin and Vallier, 2013*; *Soufi and Dalton, 2016*).

## Conserved dual effects of FGF-MAPK signaling on heart development in chordates

Previous studies highlighted how FGF-MAPK signaling is necessary along side Mesp during early cardiac development in *Ciona* (*Christiaen et al., 2008*; *Davidson, 2007*; *Davidson et al., 2006*). This early requirement also exists in vertebrates (*Abu-Issa et al., 2002*; *Alsan and Schultheiss, 2002*; *Barron et al., 2000*; *Brand, 2003*; *Reifers et al., 2000*; *Zaffran and Frasch, 2002*). We now know that these early FGF-MAPK inputs induce and maintain multipotent cardiopharyngeal states in *Ciona*, including the *Tbx1/10*+ multipotent progenitors that eventually produce the second heart lineage ([*Razy-Krajka et al., 2014*; *Stolfi et al., 2010*; *Wang et al., 2013*], [*Wang et al., 2017*] and this study). Similarly, in vertebrates, regulatory interplay between Fgf8 and Fgf10 signaling and Tbx1 is required for development of both pharyngeal arch and second heart field derivatives, presumably in part by maintaining an undifferentiated and proliferative state (*Abu-Issa et al., 2002*;

*Aggarwal et al., 2006*; *Brown et al., 2004*; *Chen et al., 2009*; *Hu et al., 2004*; *Ilagan et al., 2006*; *Kelly and Papaioannou, 2007*; *Park et al., 2006*; *Park et al., 2008*; *Vitelli et al., 2002b*; *Watanabe et al., 2010*; *Watanabe et al., 2012*). In fish, FGF signaling is necessary for cardiomyocyte addition at the arterial pole, in a manner reminiscent of its role in second heart field development (*de Pater et al., 2009*; *Lazic and Scott, 2011*). Notably, FGF signaling acts in successive phases, and its inhibition is necessary for final myocardial specification and differentiation (*Hutson et al., 2010*; *Marques et al., 2008*; *Tirosh-Finkel et al., 2010*; *van Wijk et al., 2009*). Conversely, continued FGF signaling beyond the multipotent mesodermal progenitor stages was shown to promote smooth muscle and epicardial differential in the heart (*Hutson et al., 2010*; *van Wijk et al., 2009*), and also myoblast specification and/or skeletal muscle differentiation in the head, with the expression of FGF ligands being maintained in the pharyngeal arches (*Bothe et al., 2011*; *Buckingham and Vincent, 2009*; *Michailovici et al., 2015*; *Michailovici et al., 2014*; *von Scheven et al., 2006*). Taken together, these and our data suggest that FGF-MAPK signaling plays evolutionary conserved roles during chordate cardiopharyngeal development, by promoting the specification of successive mesodermal and *Tbx1+* multipotent states, and a fate-restricted non-cardiac muscle identity, while MAPK inhibition is required for myocardial specification and differentiation in the first and second heart field, successively.

## Materials and methods

### Animals, electroporations, and chemical treatments

Gravid wild *Ciona intestinalis* type A, now called *Ciona robusta* (*Pennati et al., 2015*), were obtained from M-REP (Carlsbad, CA, USA), and kept under constant light to avoid spawning. Gametes from several animals were collected separately for in vitro cross-fertilization followed by dechorionation and electroporation as previously described (*Christiaen et al., 2009a*, *2009b*). Different quantities of plasmids were electroporated depending on the constructs. Typically, 50 µg of DNA was electroporated for NLS::lacZ or plain mCherry driving constructs but only 15 µg for *Mesp-1 >H2B::mCherry*. For perturbation constructs, 70 µg were usually electroporated, except for *Mesp > NLS::Cas9::NLS* (30 µg) and pairs of U6 >sgRNA plasmids (25 µg each). U0126 (Cell Signaling Technology, Danvers, MA) was used at 5 µM in artificial seawater from a stock solution of 20 mM in DMSO. Cytochalasin B (Sigma, Saint Louis, MO) was used at ~3 µg/mL from a 10 mg/mL stock solution in DMSO as previously performed (*Jeffery et al., 2008*). Control embryos were incubated in parallel with corresponding concentrations of DMSO alone.

### In situ hybridization

In situ hybridizations were carried out essentially as described previously (*Christiaen et al., 2009c*; *Razy-Krajka et al., 2014*), using DIG labeled riboprobes, anti-DIG-POD Fab fragments (Roche, Indianapolis, IN), and Tyramide Amplification Signal coupled to Fluorescein (Perkin Elmer, MA). Reporters expressed in the lineage of interest were marked using anti-ß-galactosidase monoclonal mouse antibody (1:1000; Promega, Fitchburg, WI) or anti-mCherry rabbit polyclonal antibody (1:500; BioVision 5993–100), respectively targeted with anti-mouse or anti-rabbit secondary antibody coupled with Alexa 648 (1:500; Invitrogen, Carlsbad, CA). The different probes used in this study were described previously (*Razy-Krajka et al., 2014*; *Stolfi et al., 2010*; *Wang et al., 2013*).

### dpERK/mcherry double fluorescent immunostaining

Samples were fixed, as for in situ hybridizations, in MEM-PFA with Tween 20 (0.05%) but only for 30 min at room temperature, washed three times in PBSt (Tween 20 0.01%) for 10 min, gradually dehydrated every 10 min in Ethanol/PBS series (33%, 50%, 80%) and Methanol 100%. Samples were then gradually rehydrated every 10 min in Methanol/PBSt series, rinsed three times in PBSt, permeabilized with PBS Triton-100 (0.2%) for 30 min and incubated for 2 hr at room temperature with anti-dpERK mouse monoclonal antibody (1:200; Sigma, Saint Louis, MO) and anti-mCherry polyclonal antibody from rabbit (1:500; Biovision, Milpitas, CA) in PBS 0.01% Triton-100 (T-Pbs) supplemented with 2% normal goat serum. Samples were then washed three times in T-PBS and incubated in anti-mouse and anti-rabbit antibodies (1 :500 each), respectively coupled with Alexa 488 and Alexa 568 (Invitrogen, Carlsbad, CA), overnight at 4°C or for 2 hr at room temperature. Finally, samples were

rinsed three times in T-PBS for 15 min and mounted in Prolong Gold (Molecular Probes, Eugene, OR).

## Molecular cloning

Coding sequences for wild-type M-Ras (KH.L172.2), Mek1/2 (KH.L147.22), Cdkn1b.a (Cdkn1b, KH.C14.564), and Cdkn1b.b (Noto16, KH.S643.6) were PCR-amplified from cDNA libraries prepared by reverse transcription of total RNA from mixed developmental stages. Insertion of the products into expressing vectors was performed using regular restriction/ligation or In-fusion (Clontech, Mountain View, CA) procedure. Oligonucleotide directed mutagenesis or two-step overlap PCRs were used to generate the point mutated forms M-Ras$^{G22V}$ and Mek$^{S220E,S216D}$ from the corresponding wild-type sequences. We also used oligonucleotide directed mutagenesis to generate mismatches in the PAM sequences adjacent to the sgRNA targets for Hand-r (153C > T 574C > T for Hand-r$^{PAMmis}$) and Tbx1/10 (325G > A and 579G > A for Tbx1/10$^{PAMmis}$). Due to the absence of a correct PAM sequence (NGG, (reverse complement CCN)), overexpressed Hand-r$^{PAMmis}$ and Tbx1/10$^{PAMmis}$ are resistant to the Cas9 nuclease activity. We also used oligonucleotide directed mutagenesis to generate the mutations in two putative Ets binding sites from the corresponding wild-type sequence of the *Tbx1/10* enhancer element: −3646TC >CT −3638GA >AG upstream from the initiation codon (E1 and E2 in *Figure 4—figure supplement 1*, respectively). Primer sequences are listed in *Supplementary file 1*.

## CRISPR/Cas9-mediated loss of Hand-r function

The pair of single guide RNA (sgRNA) targeting Tbx1/10 (sgTbx1/10) has been validated previously (*Tolkin and Christiaen, 2016*). Rescue of the Tbx1/10 loss-of-function was achieved by TVC-specific overexpression of Tbx1/10$^{PAMmis}$ driven by a *Foxf* enhancer (Foxf-1 > Tbx1/10$^{PAMmis}$). For Hand-r loss of function, sgRNAs were first designed to avoid genomic off-targets and tested as described (*Gandhi et al., 2017*). In short, sgRNA expressing cassettes (U6 > sgRNA) were assembled by single step overlap PCR. Individual PCR products (~25 µg) were electroporated with EF1a > NLS::Cas9:: NLS (30 µg), Myod905 > Venus (50 µg), driving ubiquitous expression of Cas9 and a widely expressed fluorescent reporter construct, respectively. Efficient electroporation was confirmed by observation of fluorescence before genomic DNA extraction around 16 hpf (18°C) using QIAamp DNA Micro kit (Qiagen, German Town, MD). Mutagenesis efficacy of individual sgRNAs, as a linear function of Cas9-induced indel frequency, was estimated from electrophoregrams following Singer sequencing of the targeted regions amplified from extracted genomic DNA by PCR. Result of the relative quantification of the indel frequency ('corrected peakshift' of 22% and 24%) was considered high enough for both sgRNAs targeting Hand-r, which were finally selected. The corresponding cassettes were cloned into plasmid for repeated electroporations to study the loss of function of Hand-r. Rescue of Hand-r loss-of-function was achieved by overexpression of Hand-r$^{PAMmis}$ driven by a Foxf TVC specific enhancer (Foxf-1 >Hand r$^{PAMmis}$). In order to control the specificity of the CRISPR/ Cas9 system, sgRNAs targeting *Neurogenin*, a gene not expressed in the TVC and their progeny, was electroporated in parallel. Sequences of the DNA targets and oligonucleotides used for the sgRNAs are listed in *Supplementary file 1*.

## *Tbx1/10* enhancer and *cis*-regulatory analysis

To isolate the minimal STVC-specific element of *Tbx1/10*, we used conserved non coding sequences between *Ciona robusta* and *Ciona savignyi* as a guide for molecular dissection (*Figure 4—figure supplement 1A*, http://genome.lbl.gov/vista/index.shtml; [*Frazer et al., 2004*]). We cloned a full-length *cis*-regulatory DNA construct (~7 kbp) that was analyzed by introducing large deletions to map the functional elements. We found a fragment of 1264 bp, that we called T6, located at −4682 bp upstream from the initiation codon that was sufficient for STVC expression as well as in the mesenchyme and endoderm of the reporter gene (*Figure 4—figure supplement 1A,B*). 5' deletions of the T6 element show that the main *cis*-regulatory elements required exclusively for STVC expression map in a 575 bp element, which we called T12, at −4116 bp upstream from the initiation codon sufficient (*Figure 4—figure supplement 1A*). The sequence of this element, which we called *T12*, reveals the presence of putative Ets1/2 binding sites (*Figure 4—figure supplement 1C*) that were predicted using JASPAR (*Khan et al., 2018*) and CisBP (*Weirauch et al., 2014*) databases. The

minimal *Tbx1/10* STVC-specific enhancer was further analyzed using point mutations of the candidate Ets1/2 sites with highest predicted scores (*Figure 4—figure supplement 1D*).

## Observation and imaging

Samples were usually scored under a DM2500 epifluorescent microscope (Leica Microsystems, Wetzlar, Germany). Imaging was performed using a TCS SP8 X inverted confocal microscope equipped with a white light laser, AOBS and HyD detectors (Leica Microsystems).

## Acknowledgement

We thank Robert Kelly (Université Aix-Marseille, CNRS, France) for feedbacks on the manuscript. We are grateful to Alberto Stolfi for collaborative inputs throughout the project. We thank Farhana Salek and Kristyn Millan for technical support. This project was funded by NIH/NHLBI R01 award HL108643, and trans-Atlantic network of excellence award 15CVD01 from the Leducq Foundation to LC.

## Additional information

### Funding

| Funder | Grant reference number | Author |
|---|---|---|
| National Heart, Lung, and Blood Institute | HL108643 | Lionel Christiaen |
| Fondation Leducq | 15CVD01 | Lionel Christiaen |
| AFM-Téléthon | 20895 | Basile Gravez |

The funders had no role in study design, data collection and interpretation, or the decision to submit the work for publication.

### Author contributions

Florian Razy-Krajka, Conceptualization, Data curation, Formal analysis, Validation, Investigation, Visualization, Methodology; Basile Gravez, Data curation, Formal analysis, Investigation, Visualization; Nicole Kaplan, Claudia Racioppi, Wei Wang, Formal analysis, Investigation, Visualization; Lionel Christiaen, Conceptualization, Resources, Supervision, Funding acquisition, Validation, Writing—original draft, Project administration, Writing—review and editing

### Author ORCIDs

Lionel Christiaen  https://orcid.org/0000-0001-5930-5667

### Decision letter and Author response

Decision letter https://doi.org/10.7554/eLife.29656.019
Author response https://doi.org/10.7554/eLife.29656.020

## Additional files

### Supplementary files

• Supplementary file 1. Primer sequences.
DOI: https://doi.org/10.7554/eLife.29656.016

• Transparent reporting form
DOI: https://doi.org/10.7554/eLife.29656.017

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
