## [Decision Letter]

Thank you for submitting your article "An FGF-driven feed-forward circuit for spatiotemporal patterning of the cardiopharyngeal mesoderm in a simple chordate" for consideration by *eLife*. Your article has been reviewed by three peer reviewers, and the evaluation has been overseen by Marianne Bronner as the Senior and Reviewing Editor. The following individual involved in review of your submission has agreed to reveal his identity: Nori Satoh (Reviewer #2).

The reviewers have discussed the reviews with one another and the Reviewing Editor has drafted this decision to help you prepare a revised submission.

Summary:

In Ciona embryos, the B7.5 progenitor gives rise to larval anterior tail muscle progenitors, as well as to progenitors of the juvenile heart and atrial siphon muscles (ASM). In this manuscript, the authors present a detailed dissection of the gene regulatory networks underlying the formation of the non-cardiogenic ASM derivatives of the B7.5. This study describes the spatial and temporal epistatic relationships between FGF signalling and *Hand-r, Tbx1/10* and *Ebf* transcription factors. It also covers the importance of cell divisions in the establishment of the ASM program. This is a very solid work, which will attract the attention of the ascidian community and also suggests some interesting experiments to the community of mammalian scientists interested in heart development.

Essential revisions:

1) The authors need to clarify changes in cell identity following genetic perturbations. What are the cell identities of the cells with loss of FGF/MAPK signaling, especially in the late perturbations? Does the loss of *Ebf* expression lead to SHP identity by default? Existing markers for the other lineages, and in particular for the heart progenitors would be appropriate. They need to clarify if cells simply are making binary decisions within the cardiopharyngeal lineages or if fate specification a more sophisticated process.

2) It is interesting that the patterns of cell division (orientation and inequality) are independent of the perturbations introduced by the authors. This suggests a possible uncoupling of fate and cell behaviors. This should be better quantified. A more thorough description of cell behaviors in perturbed conditions would strengthen the manuscript and clarify if there is uncoupling between fate specification and unequal division.

3) What is the evidence that the delayed activation of ASM marker gene expression observed is not sufficient to drive ASM formation? Is the timing of division altered?

4) Could interference with FGFR, *Hand-r, Tbx1/10* or *Ebf* affect the activity of the *FoxF* enhancer used to drive M-ras, MEK, *Tbx1/10* or *Ebf*, or more generally the amount of these proteins in epistatic experiments?

5) The authors show that *Hand-r* expression is inhibited upon dnFGFR expression in late TVCs. Does this mean that in early TVCs *Hand-r* expression is detectable? Are TVCs correctly specified in this experimental set-up?

6) The claims that the asymmetric divisions are unperturbed are not supported by presented data. On what basis are the larger daughter cells identified?

---

## [Author Response]

Essential revisions:1) The authors need to clarify changes in cell identity following genetic perturbations. What are the cell identities of the cells with loss of FGF/MAPK signaling, especially in the late perturbations? Does the loss of Ebf expression lead to SHP identity by default? Existing markers for the other lineages, and in particular for the heart progenitors would be appropriate. They need to clarify if cells simply are making binary decisions within the cardiopharyngeal lineages or if fate specification a more sophisticated process.

This important point is addressed experimentally in our companion paper currently available on the BioRxiv and which we are revising for publication in another peer-review journal. From that standpoint, the data concerning the impact of perturbations on heart gene expression is published in BioRxiv or will be published in this paper.

Briefly, in this paper we used FACS and bulk RNA-seq to profile the transcriptional effects of *Foxf>dnFGFR* and *Hand-r>caMRas* at 12, 15, 18 and 20hpf and showed that FGF-MAPK basically toggled between heart- and ASM-specific programs.

In the revised version, we validate these data by FISH focused on pan-cardiac, SHP- and FHP-specific markers including *Lrp4/8, Kirr* and *Dach* showed in Figure 5. Specifically, we studied the novel SHP marker *Dach* more in-depth and showed that its expression requires both *Tbx1/10* and termination of FGF-MAPK signaling.

2) It is interesting that the patterns of cell division (orientation and inequality) are independent of the perturbations introduced by the authors. This suggests a possible uncoupling of fate and cell behaviors. This should be better quantified. A more thorough description of cell behaviors in perturbed conditions would strengthen the manuscript and clarify if there is uncoupling between fate specification and unequal division.

This is an important question, which we addressed and reported the effects in novel Figure 2—figure supplement 1. The most important aspect for this paper is that, although FGF-MAPK does impact cell divisions patterns, the effects on cell divisions do not account for the observed effects on gene expression, which is the focus of this study.

A full characterization of the relationship between FGF-MAPK signaling and asymmetric and oriented cell divisions is to some extend outside the scope of this study, and more directly part of another study, which we are in the process of completing for a separate paper.

3) What is the evidence that the delayed activation of ASM marker gene expression observed is not sufficient to drive ASM formation? Is the timing of division altered?

We added new data (Figure 5) to show that the main effect of delayed STVC division is to eliminate the SHP lineage. To clarify, the problem is not ASM specification, which still occurs as shown by eventual *Ebf* upregulation by 18hpf, instead it is potentially abolishing SHP specification.

In a way, we agree that the entrainment of *Ebf* by cell divisions may be dispensable for ASM specification, and mostly revealed that the endogenous dynamics of the feedforward circuit is slower than observed in the embryo. We proposed that this coupling with the cell cycle ensures that cell division, and restriction of the FGF-MAPK signaling, occurs prior to *Tbx1/10* and *Ebf* upregulation, and contributes to the timely onset of gene expression.

In the paper, we used the delayed induced by Wee1 to further probe the switch from an early MAPK-dependent to a MAPK-independent mode of *Ebf* regulation, which we propose is the transition from early -reversible- specification to commitment to an ASM identity.

Arguably, the cell cycle control on gene expression is somewhat preliminary, as we acknowledge in the Discussion, and definitely worth pursuing (Gravez et al., ongoing research).

4) Could interference with FGFR, Hand-r, Tbx1/10 or Ebf affect the activity of the FoxF enhancer used to drive M-ras, MEK, Tbx1/10 or Ebf, or more generally the amount of these proteins in epistatic experiments?5) The authors show that Hand-r expression is inhibited upon dnFGFR expression in late TVCs. Does this mean that in early TVCs Hand-r expression is detectable? Are TVCs correctly specified in this experimental set-up?

These points 4 and 5 are important controls. We initially surmised that the perturbations would not markedly affect early TVC gene expression because it takes ~2 or more hours for *Foxf*-driven gene products to accumulate and have a detectable effect (as seen for GFP expression for example). In line with this idea, we generated novel supplementary data presented in Figure 1—figure supplement 2, which indicates that the defined perturbations do not affect early TVC induction, nor the expression of *Foxf*-driven transgenes.

6) The claims that the asymmetric divisions are unperturbed are not supported by presented data. On what basis are the larger daughter cells identified?

This question is somewhat similar to point #2, which we addressed experimentally in novel Figure 1—figure supplement 2.